# A comprehensive data-driven model of cat primary visual cortex

Ján Antolík[1,2,3☯]*, Rémy Cagnol[1☯], Tibor Rózsa[1], Cyril Monier[2,4], Yves Frégnac[2,4], Andrew P. Davison[2,4]

**1** Faculty of Mathematics and Physics, Charles University, Malostranské nám. 25, Prague 1, Czechia, **2** Unit of Neuroscience, Information and Complexity (UNIC), CNRS FRE 3693, Gif-sur-Yvette, France, **3** INSERM UMRI S 968; Sorbonne Université, UPMC Univ Paris 06, UMR S 968; CNRS, UMR 7210, Institut de la Vision, Paris, France, **4** Institut des neurosciences Paris-Saclay, Université Paris-Saclay, CNRS, Saclay, France

☯ These authors contributed equally to this work.
* antolikjan@gmail.com

**Data Availability Statement:** The Mozaik python library, in which our primary visual cortex model is implemented, is freely accessible here: https://github.com/CSNG-MFF/mozaik All the parameters for this specific model instance, as well as the code for running all the experiments and analyses

## Abstract

Knowledge integration based on the relationship between structure and function of the neural substrate is one of the main targets of neuroinformatics and data-driven computational modeling. However, the multiplicity of data sources, the diversity of benchmarks, the mixing of observables of different natures, and the necessity of a long-term, systematic approach make such a task challenging. Here we present a first snapshot of a long-term integrative modeling program designed to address this issue in the domain of the visual system: a comprehensive spiking model of cat primary visual cortex. The presented model satisfies an extensive range of anatomical, statistical and functional constraints under a wide range of visual input statistics. In the presence of physiological levels of tonic stochastic bombardment by spontaneous thalamic activity, the modeled cortical reverberations self-generate a sparse asynchronous ongoing activity that quantitatively matches a range of experimentally measured statistics. When integrating feed-forward drive elicited by a high diversity of visual contexts, the simulated network produces a realistic, quantitatively accurate interplay between visually evoked excitatory and inhibitory conductances; contrast-invariant orientation-tuning width; center surround interactions; and stimulus-dependent changes in the precision of the neural code. This integrative model offers insights into how the studied properties interact, contributing to a better understanding of visual cortical dynamics. It provides a basis for future development towards a comprehensive model of low-level perception.

## Author summary

Computational modeling can integrate fragments of understanding generated by experimental neuroscience. However, most models considered only a few features of neural computation at a time, leading to either poorly constrained models with many parameters, or lack of expressiveness in over-simplified models. A solution is to develop detailed models, but constrain them with a broad range of anatomical and functional data to

performed in this manuscript is available here: https://github.com/CSNG-MFF/mozaik-models/ tree/main/LSV1M The results shown in this manuscript, as well as the specific models and experiments parameters are available here: http:// v1model.arkheia.org/simruns.

**Funding:** JA,CM,YF and AD were supported by the Centre National de la Recherche Scientifique, Paris-Saclay IDEX (NeuroSaclay); the French National Research Agency (Complex-V1); the European Union's Seventh Framework Program under grant agreements 269921 (BrainScaleS) and 604102 (Human Brain Project); JA, RC and TR were supported through the project Improvement of internationalization in the field of research and development (CZ.02.2.69/0.0/0.0/17_050/ 0008466) at Charles University, through institutional funding from Charles University (projects PRIMUS/20/MED/006 and 1179 GAUK/ 4089/2022). JA and RC were also supported through ERDF-Project Brain dynamics (CZ.02.01.01/00/22_008/0004643). The funders had no role in study design, data collection and analysis, decision to publish, or preparation of the manuscript.

**Competing interests:** The authors have declared that no competing interests exist.

prevent overfitting. This requires a long-term systematic approach. Here we present a first snapshot of such an integrative program: a large-scale spiking model of cat primary visual cortex, that is constrained by an extensive range of anatomical and functional features. Together with the associated modeling infrastructure, this study lays the groundwork for a broad integrative modeling program seeking an in-depth understanding of vision.

# 1 Introduction

A key challenge in neuroscience is the consolidation of the multitude of experimental findings into a coherent characterization of cortical processing. Slow progress in such knowledge integration is evidenced by the surprisingly incomplete state of understanding even in well-explored cortical areas, such as the primary visual cortex (V1) [1].

For example, while V1 circuitry supports many different computations occurring concurrently, such as edge detection, depth processing, and contextual modulation, these have mostly been studied in isolation. Many computational studies have proposed mechanisms to explain V1 phenomena—one at a time—including layer-specific resting-state differences [2], contrast adaptation [3], or orientation tuning contrast invariance [4–8]. An alternative to such an unipotent model approach is to propose a common adaptive mechanism capable of explaining multiple V1 properties (e.g. [9, 10]). However, only few concurrent phenomena have been actually demonstrated in such models (but see [11]), despite the potential these approaches hold for knowledge integration. The most comprehensive explanations of V1 function come from a few large-scale models combining well-established cortical mechanisms [12, 13], but these studies still account for only a limited number of phenomena at a time (Section 3.3).

Failure to address the full spectrum of phenomena concurrently in a single model instance leads to under-constrained models, and proliferation of numerous alternative explanations for individual phenomena, while informing us little about how different V1 computations are multiplexed such that the same synapses participate in multiple simultaneous calculations. While further reductionist, hypothesis-led research is indisputably required, many of these questions require a systematic, integrative approach to progressively build a unified multi-scale theory of brain function [14, 15]. A danger in detailed modeling is growth of free parameters, risking over-fitting and consequent lack of explanatory power. However, building and validating a model against a large number of diverse experimental studies adds strong constraints on parameters. Similarly, a multi-scale approach greatly increases the categories of experiments suitable as a source of model constraints [16].

In our view, the solution to these challenges is a sustained, collaborative effort to incorporate the full breadth of experimentally established constraints into a single pluripotent model of V1. While necessarily a long-term project, here we offer a basis for such an integrationist effort: a detailed, biologically realistic model of anesthetized cat V1, validated against an unprecedented range of experimental measures, including intra- and extra-cellular functional recordings realized over the years in our laboratory. To support collaborative knowledge integration, the model and all experimental protocols are implemented in an easily extensible simulation framework [17]. The complete model's code is published in an online repository (Section 4), and all model parameters, stimuli and experimental protocols along with the analyzed results can also be explored online (Section 3.1).

The model comprises cortical layers 4 and 2/3, within a $5.0 \times 5.0$mm patch around the area centralis (Fig 1; see Methods). Afferent connections to layer 4 neurons are stochastically generated based on receptive-field templates parametrised by the neurons' retinotopic positions,

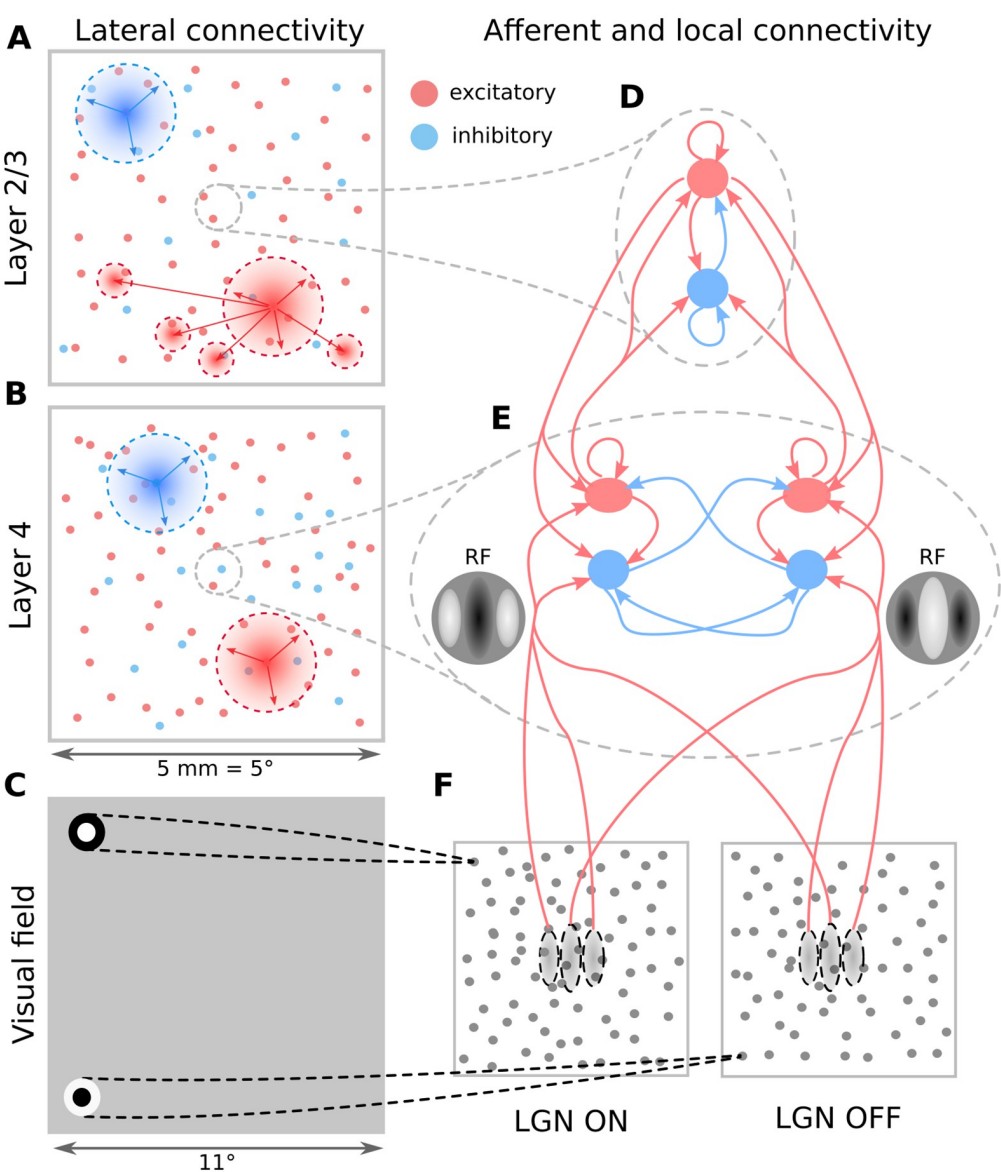

**Fig 1. The model architecture.** (A-B) Layer 2/3 and Layer 4 lateral connectivity. All cortical neuron types make local connections within their layer. Layer 2/3 excitatory neurons also make long-range functionally specific connections. For the sake of clarity A,B do not show the functional specificity of local connections and connection ranges are not to scale. (C) Extent of modeled visual field and example of receptive fields (RFs) of one ON and one OFF-center lateral geniculate nucleus (LGN) relay neuron. As indicated, the model is retinotopically organized. The extent of the modeled visual field is larger than the corresponding visuotopic area of modeled cortex in order to prevent clipping of LGN RFs. (D) Local connectivity scheme in Layer 2/3: connections are orientation- but not phase-specific, leading to predominantly Complex cell type RFs. Both neuron types receive narrow connections from Layer 4 excitatory neurons. (E) Local connectivity in Layer 4 follows a push-pull organization. (F) Afferent RFs of Layer 4 neurons are formed by sampling synapses from a probability distribution defined by a Gabor function overlaid on the ON and OFF LGN sheets, where positive parts of the Gabor function are overlaid on ON and negative on OFF-center sheets. The ON regions of RFs are shown in white, OFF regions in black.

and their location in an overlaid, pre-computed orientation map (Fig 2). Intra-laminar connectivity in Layer 4 follows a push-pull organization [5] (Fig 1E). The recurrent excitatory connectivity in Layer 2/3 has both a distance-dependent bias and a functional bias based on the orientation selectivity of the neurons [18]. The distance-dependent intra-cortical connectivity

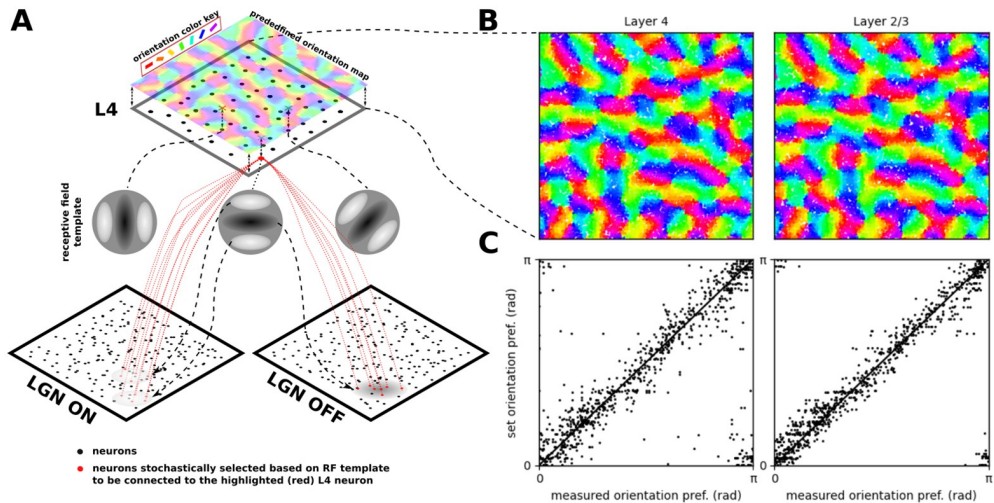

**Fig 2. The induction of orientation maps in the model.** (A) The precomputed orientation map that serves as a template based on which orientation of the thalamo-cortical connectivity of Layer 4 neurons is determined. (B) The orientation assigned to neurons (each dot represents a neuron) in the two layers based on their position within the overlaid pre-computed orientation map. (C) The comparison between the orientation preference assigned to the neurons via the pre-computed orientation map (y axis) and the orientation preference measured using the sinusoidal grating protocol (x axis; Section 4.4).

follows parameters derived from experimental data [19, 20] (Fig 1A and 1B). The model has been probed by the most common stimulation paradigms used in early vision: drifting gratings (DGs), masked gratings of variable diameter, and naturalistic movies. Under background thalamic bombardment, a spontaneous cortical activity within a physiologically plausible resting-conductance regime emerges in the model. Under visual stimulation, the same model exhibits diverse patterns of interplay between evoked excitation and inhibition; stimulus-locked sub-threshold variability; contrast-invariant orientation tuning; size tuning; stimulus-dependent firing precision; and a realistic distribution of Simple/Complex receptive fields.

The primary contribution of this study is that of knowledge integration: we advance our ability to capture the function of V1 within a single model, and provide insights into how the studied properties interact, contributing to a better understanding of cortical dynamics. We demonstrate for the first time in a model of visual system a self-consistent spontaneous state without unrealistic sources of variability, which allows us to reproduce stimulus dependent changes in trial-to-trial variability of V1 responses previously demonstrated experimentally [21]. To the best of our knowledge, we are the first to demonstrate how a model of V1 can achieve orientation tuning of firing rate, membrane-potential, and excitatory & inhibitory conductance simultaneously (in line with experimental evidence), which is an important constraint on the cortical dynamics driving generation of orientation tuning in V1. We demonstrate the utility of the proposed function-driven integrative approach and provide a basis for future development towards a comprehensive model of V1 and beyond, grounded in an open-science approach.

## 2 Results

Here we present a single model instance (i.e. a single set of equations with a single set of parameters was used for all simulations; only the visual stimuli changed) representing the cat

early visual system, consisting of a retino-thalamic pathway feeding input in the form of spikes to layers 4 and 2/3 of a 5.0 × 5.0 mm patch of V1 (Figs 1 and 2) centered at 3 degrees of visual field eccentricity (see Methods).

## 2.1 Spontaneous activity

The only external source of variability in the model is a white noise current injection into lateral geniculate nucleus (LGN) cells which induces their spontaneous firing at about 17 sp/s for ON cells and 8 sp/s for OFF cells, in line with recordings from LGN in response to a gray blank stimulus at 50 cd/m$^2$ [22]. This noise represents the combined effect of the retinal spontaneous discharge [23–27], and the intrinsic mechanisms generating ongoing activity in thalamus [28]. Due to the low proportion of synapses of thalamic origin in the model cortex [29], the variability of spontaneous and visually evoked cortical activity is largely driven by internal cortical dynamics (S1 Fig).

In the spontaneous condition all modeled excitatory neuron populations fire irregularly (Figs 3A and 4B: coefficient of variation of their inter-spike-intervals is above 0.9) and asynchronously (Figs 3A and 4C: the mean cross-correlation of the spike counts is less than 0.01), with only occasional synchronized events, consistent with spontaneous firing patterns in

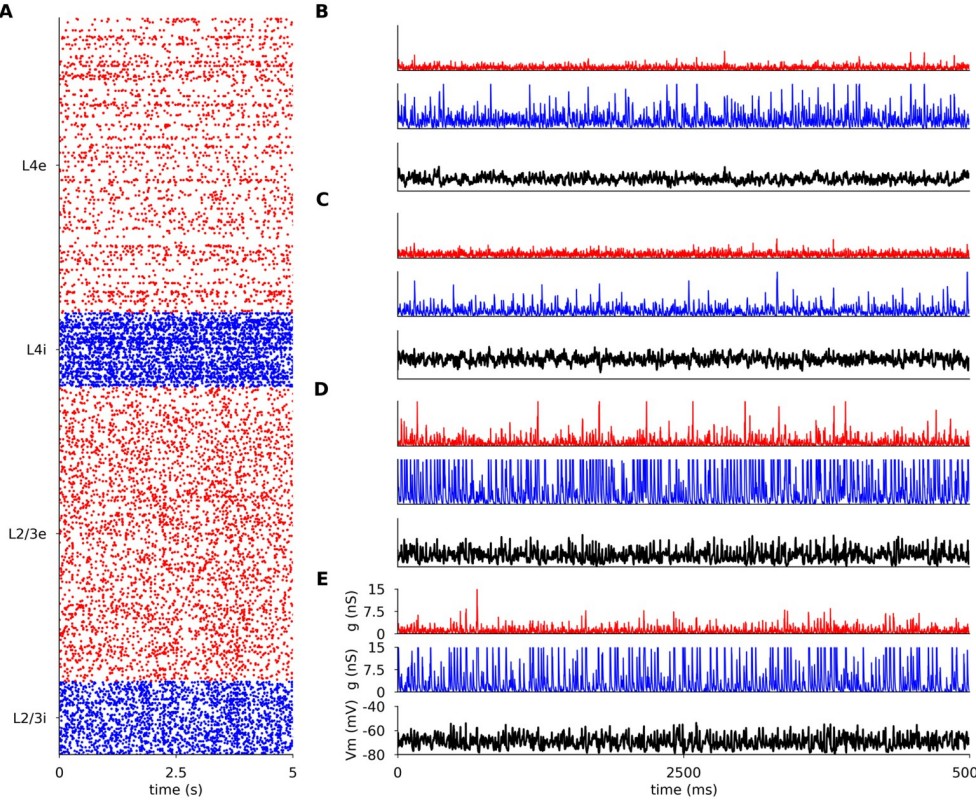

**Fig 3. Spontaneous activity in modeled cortical layers.** (A) Raster plot of spiking activity recorded for 5 s of biological time in Layers 4 and 2/3 (from bottom to top; red: excitatory neurons, blue: inhibitory neurons). The relative number of spike trains shown corresponds to the relative number of neurons in the network (total of 1000 neurons shown). (B-E) the membrane potential (black) and the excitatory (red) and inhibitory (blue) synaptic conductances of randomly selected neurons. (B) excitatory neuron in Layer 2/3; (C) inhibitory neuron in Layer 2/3 (C); (D) excitatory neuron in Layer 4; (E) inhibitory neuron in Layer 4.

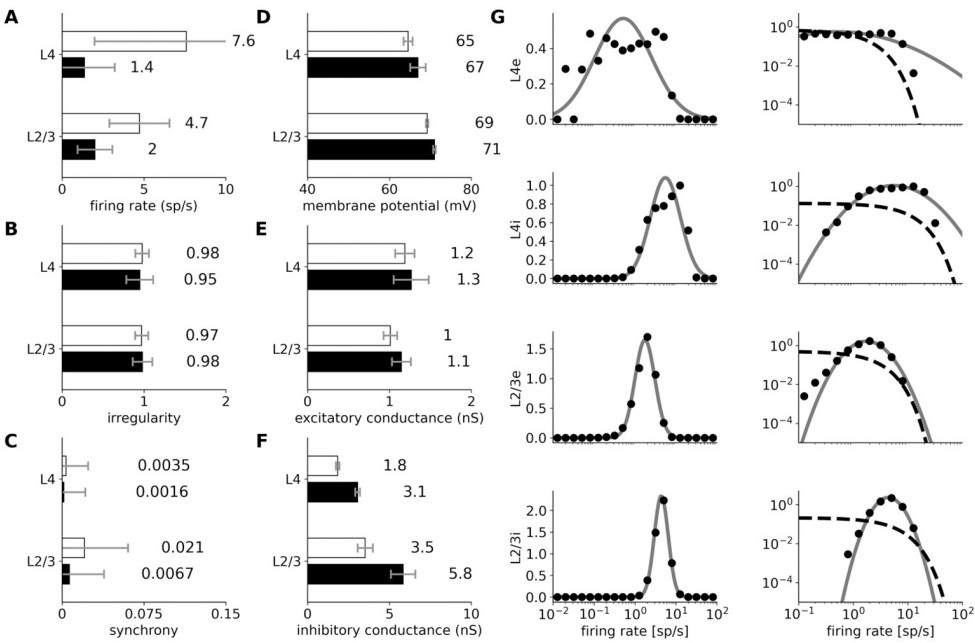

**Fig 4. Statistics of spontaneous activity in the simulated network recorded for 40 seconds.** (A–F) Six measures of spontaneous activity for excitatory (black bars) and inhibitory (white bars) neural populations in the two modeled cortical layers. Error bars report standard deviation. (A–C) Statistics of the spiking activity of the 4 modeled cortical populations based on spike trains recorded for 40 s. (A) Mean single-unit firing rates. (B) Irregularity of single-unit spike trains quantified by the coefficient of variation of the inter-spike intervals. (C) Synchrony of multi-unit spiking activity quantified as the mean correlation coefficient between the spike histogram (bin size 10 ms) of all pairs of recorded neurons in a given population. (D–F) Mean values of intracellular variables: membrane potential (D); excitatory (E) and inhibitory (F) synaptic conductances. (G) The distribution of firing rates (black points) in Layer 2/3 excitatory and inhibitory neurons is well fitted by a log-normal distribution with matching mean and variance (gray line; left-column). The log-normal distribution appears as a normal distribution on a semi-logarithmic scale. The data are better fit by a log-normal (gray line) than by an exponential (dashed line) distribution (right-column; log-log scale).

awake animals [30, 31] and with the up-states in *in vivo* anesthetized preparations [30, 32]. The model exhibits higher spontaneous correlations between inhibitory than between excitatory neurons (Figs 3A and 4C), which is consistent with the observation of higher synchrony among inhibitory neurons in macaque cortex during slow wave sleep [33]. Overall the spontaneous activity corresponds to the asynchronous irregular state previously observed in balanced randomly connected network models [34–36].

We observe higher mean spontaneous rates among inhibitory than excitatory neurons, in line with experimental evidence [37, 38]. The spontaneous firing rates of the majority of excitatory neurons are low (<2 sp/s, Fig 4A and 4G), as observed experimentally [39, 40]. In addition, higher spontaneous firing rates in Layer 2/3 than in Layer 4 have been reported in cat [41], just like in the model. In both excitatory and inhibitory populations of Layer 2/3, the firing rates closely follow log-normal distributions (Fig 4G), a phenomenon previously shown in several cortical areas and species [42, 43].

The mean resting membrane potential (Vm) of neurons is close to -70 mV, as observed in *in vivo* cat V1 recordings [44]. As shown in Figs 3B, 4E and 4F, the excitatory synaptic conductances recorded in model neurons during spontaneous activity (1.15 nS, averaged across all layers) are consistent with levels observed in cat V1 ($\sim$ 1 nS; layer origin not known) [44]. The inhibitory conductance during spontaneous activity was 3.56 nS when averaged across all

model neurons, slightly lower than the value of 4.9 nS reported in cat [44]. These realistic levels of conductance are in stark contrast to those in most balanced random network models of spontaneous activity, in which the global synaptic conductance tends to be orders of magnitude higher [36, 45]. Overall, the model exhibits a spontaneous state with balanced excitatory and inhibitory inputs well matched with cat V1 at both extra- and intracellular level, without the need for any unrealistic external sources of variability. This represents an advance over previous models of V1 and over dedicated models of spontaneous activity, and enables a self-consistent exploration of stimulus-driven changes in variability (Section 2.4).

## 2.2 Responses to drifting sinusoidal gratings

Fig 5 shows the responses of representative excitatory and inhibitory neurons from both model cortical layers to multiple trials of optimally- and cross-oriented sinusoidal gratings of optimal spatial frequency, drifting at 2 Hz.

The membrane potential of the Layer 4 excitatory cells follow the sinusoidal modulation of the grating luminance—characteristic of Simple cells [44]—but remains largely unmodulated and below threshold during the cross-oriented condition (Fig 5A (bottom panel)). The sinusoidal shape of the membrane potential trace is the result of the excitatory and inhibitory synaptic conductances being mutually in anti-phase (Fig 5A middle panel), due to the push-pull connectivity in model Layer 4. The trial-averaged excitatory and inhibitory synaptic conductances (mean value 1.66 nS and 4.15 nS respectively) are consistent with observations in cat V1 [21, 44]. In layer 4 the spikes are generated in a periodic manner aligned to the upswings of the membrane potential, while the cross-oriented grating does not elicit significant depolarization (Fig 5A (top panel)).

The spiking response of the inhibitory cells in Layer 4 (Fig 5B) is similar. The smaller number of excitatory cortico-cortical synapses is compensated by the stronger excitatory to inhibitory synaptic weights (see Methods), leading to comparable mean input excitatory conductances (1.55 nS). However, the smaller number of inhibitory cortico-cortical synapses

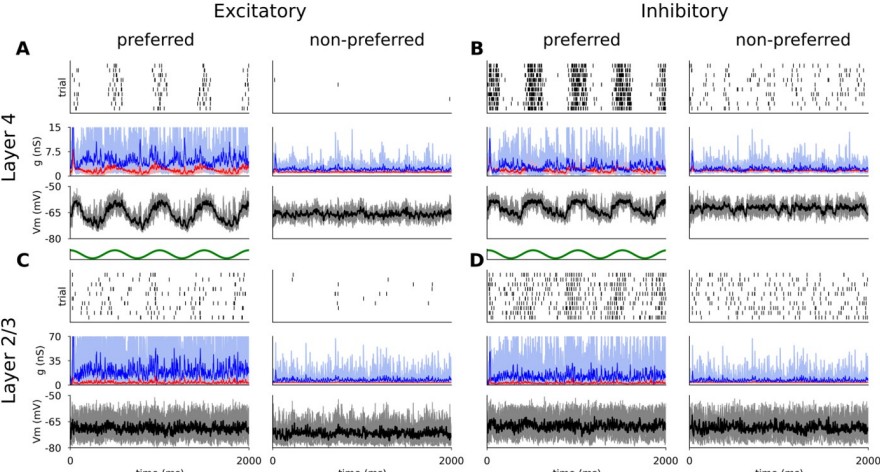

**Fig 5. Responses of representative excitatory (A,C) and inhibitory (B,D) neurons from Layer 4 (A,B) and Layer 2/3 (C,D) to 10 trials of DGs of optimal (left) and cross-oriented (right) orientations.** For all neurons, the top panel is a spike raster plot, each line corresponding to a single trial. The middle panel shows the incoming excitatory (red) and inhibitory (blue) synaptic conductances and the bottom panel shows the membrane potential (thin lines: single trials; thick lines: mean). Stimulus onset is at time zero.

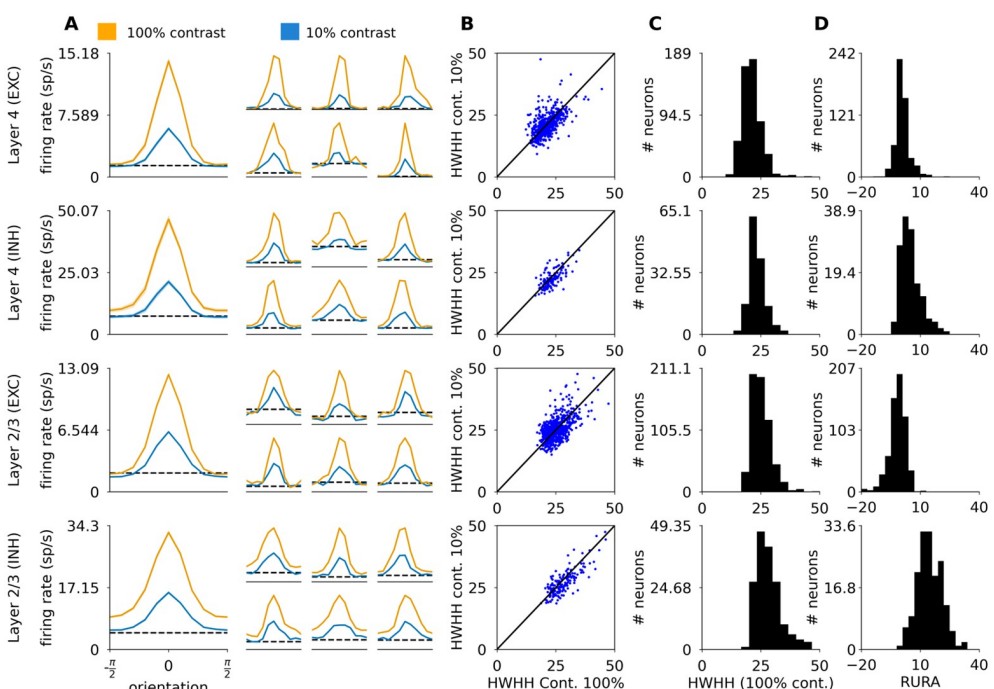

**Fig 6. Orientation tuning properties of excitatory and inhibitory neurons in Layers 4 and 2/3.** (A) Orientation tuning curves calculated as the mean firing rate of the neuron at the given orientation. The large tuning curves on the left represent the mean of centered orientation tuning curves across all the measured neurons of the given type. The smaller tuning curves on the right correspond to randomly selected single neurons. The shaded areas correspond to the standard error of the mean. (B) Half-width-at-half-height (HWHH) at 100% vs 10% contrast. (C) The distribution of HWHH measured at 100% contrast across the four modeled cortical populations. (D) The distribution of relative unselective response amplitude (RURA) measured at 100% contrast across the four modeled cortical populations.

yields lower mean input inhibitory conductances (2.65 nS), resulting in a shift towards stronger relative excitation during visual stimulation, and hence higher overall firing rates (Section 2.2.1). Furthermore, due to the weaker inhibition, the cross-oriented grating still elicits slight depolarization of the membrane potential, leading to increased spiking activity, and resulting in broadening of the orientation tuning curve, as further examined in Section 2.2.1.

Unlike the phasic response of Layer 4 cells, both excitatory and inhibitory cells in Layer 2/3 show steady depolarization in response to an optimally orientated grating (Fig 5C and 5D), similar to non-linear Complex cells [46, 47]. The mean excitatory and inhibitory synaptic conductances (2.48 nS and 14.01 nS respectively) are higher than in Layer 4 due to the higher number of recurrent connections (see Methods), and lie within the ranges observed experimentally [21, 44].

**2.2.1 Orientation tuning of spike response.** We next examined the responses of the model to different orientations of the sinusoidal grating at two different contrasts (Fig 6A and 6B). The mean tuning widths of excitatory and inhibitory neurons in Layer 4 and excitatory and inhibitory neurons in Layer 2/3, measured as half-width-at-half-height (HWHH), were 21.1˚, 23.3˚, 25.2˚ and 28.3˚ respectively (Fig 6C), which is within the range of tuning widths in cat V1 reported by experimental studies [8, 48, 49]. Even though some sub-groups of inhibitory neurons have been shown to be broadly tuned to orientation [48], on average inhibitory neurons are well tuned [48, 49], just as in our model. Indeed, Nowak *et al.* [48] found that inhibitory layers are slightly more broadly tuned than excitatory ones (mean HWHH of

                                    

regular spiking 23.4˚ vs. fast spiking neurons 31.9˚). Our model follows the same trend (mean HWHH of excitatory neurons 23.4˚ vs. inhibitory neurons 26.0˚).

Because the HWHH measure does not take into account the unselective portion of the response, cells that respond substantially above the spontaneous rate at all orientations, but where this unselective response is topped by a narrow peak, will yield low HWHH. To address this concern, Nowak *et al.* [48] also calculated the relative unselective response amplitude (RURA; see Methods), and showed that the RURA of inhibitory neurons is considerably broader than that of excitatory ones (mean for regularly spiking neurons 2.5% vs. fast spiking neurons neurons 18.1%). We have repeated this analysis, and found that the difference between the RURA of excitatory and inhibitory model neurons (Fig 6C and 6D; -0.47% in excitatory vs 10.43% in inhibitory neurons) is indeed more pronounced than their difference in HWHH.

Most excitatory and inhibitory neurons in the model cortical Layer 4 and 2/3 exhibit contrast invariance of orientation tuning width (Fig 6A and 6B). On average we observe very minor broadening of the tuning curves at high contrast: the mean HWHH differences between low and high contrasts is 0.19˚ (excitatory, Layer 4), 0.75˚ (inhibitory, Layer 4), −0.05˚ (excitatory, Layer 2/3), and 1.06˚ (inhibitory, Layer 2/3). Such minor broadening has been observed experimentally in Simple cells in cat V1 (0.3˚; layer origin and neural type not known [8]).

Overall, the orientation tuning in our model is in good quantitative agreement with experimental data, with both excitatory and inhibitory neurons exhibiting sharp and contrast-invariant orientation tuning across all modeled layers, in contrast to many previous modeling studies that relied on untuned inhibition, or broad contrast-dependent inhibition [5, 6, 50]. See the Discussion section 3.4 for a mechanistic explanation of how these phenomena arise in this model in comparison to others.

**2.2.2 Orientation tuning of sub-threshold signals.**   To better understand the mechanisms of orientation tuning in the model, we investigated the tuning of the membrane potential and the excitatory and inhibitory synaptic conductances. An essential mechanism responsible for orientation tuning in this model is the push-pull connectivity bias in Layer 4. In theory, a cell with a pure push-pull mechanism with perfectly balanced excitation and inhibition should exhibit perfectly linear (Simple cell) behavior, where orientation tuning is solely driven by the luminance modulation of the DG, and the mean membrane potential remains at rest at all orientations. To assess to what extent this idealized scheme is true in this model, we have plotted the orientation tuning curves of the mean (F0) and first harmonic (F1) components of the sub-threshold signals across the different neural types and layers in Fig 7.

In Layer 4 the orientation tuning of the membrane potential is dominated by the F1 component, in line with the presence of strong push-pull mechanisms in this model layer (Fig 7B). Layer 4 cells also have significant tuning of the F0 component (mean membrane potential), but of lesser magnitude (Fig 7A). The F0 components of the membrane potential show noticeable broadening at higher contrast, consistent with experimental observations that, unlike the spiking response, the orientation tuning width of the mean of membrane potential is not contrast independent [8]. In Layer 2/3 the magnitude of the membrane potential components is reversed: the magnitude of the F0 component of the membrane potential is stronger than the F1 component. This is consistent with the lack of phase specific connectivity in Layer 2/3, and leads to the predominance of Complex cells in this model layer (Section 2.2.3).

To understand how the interplay between excitation and inhibition leads to the observed membrane potential tuning characteristics, we plot the orientation tuning curves of the F0 and F1 components of excitatory (Fig 7C and 7D) and inhibitory (Fig 7E and 7F) synaptic conductances. In Layer 4, the most obvious difference is that, in comparison to the tuning of membrane potential, the F0 component of both excitatory and inhibitory conductances is not

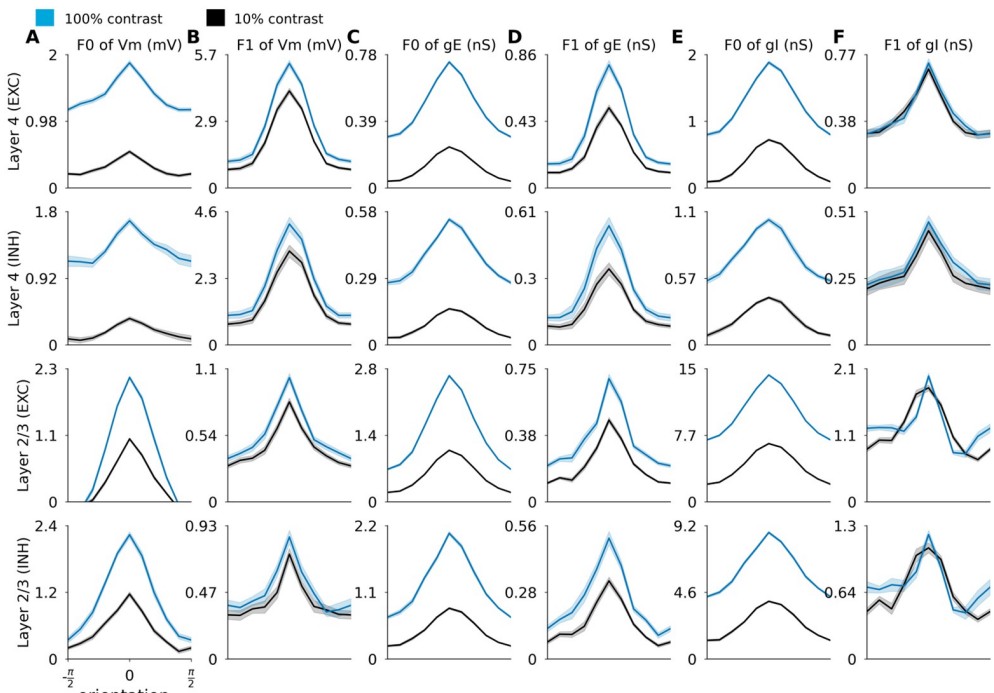

**Fig 7. Orientation tuning of membrane potential and excitatory and inhibitory conductances of excitatory and inhibitory neurons in Layer 4 and 2/3.** Each row shows the average orientation tuning curves of all neurons of given neural type recorded in the given layer (see row labels on the left), with spontaneous level of each respective neural signal subtracted. Orientation tuning curves of: (A) F0 component of membrane potential; (B) F1 component of membrane potential; (C) F0 component of excitatory synaptic conductance; (D) F1 component of excitatory synaptic conductance; (E) F0 component of inhibitory synaptic conductance; (F) F1 component of inhibitory synaptic conductance. Low contrast: 10% (black line), high contrast: 100% (blue line). The shaded areas correspond to the standard error of the mean.

dominated by the F1 component. The weaker F0 component of the membrane potential in Layer 4 neurons is thus due to the cancellation between the excitatory and inhibitory F0 components, while the F1 component of the membrane potential orientation tuning remains strong, as cancellation between excitatory and inhibitory conductances does not occur due to the half-period phase shift between them (see Fig 5A and 5B). In Layer 2/3 both F0 and F1 components of excitatory and inhibitory conductances in both excitatory and inhibitory neurons are well tuned, but the F0 components dominate in magnitude the F1 components for both excitatory and inhibitory conductances and for both neural types.

To the best of our knowledge, this is the first model where orientation tuning of both F0 and F1 components of membrane potential and conductances is exhibited in all layers and cell types, consistent with experimental observations [51, 52]. The contrast invariance in orientation tuning arises as a complex interplay between excitation and inhibition, differs between layers and neural types, and, in the model Layer 4, is consistent with engagement of a push-pull mechanism [5]. However, additional enhancement of activity is present due to the recurrent facilitatory dynamics among neurons of similar orientation, which goes beyond the predictions of the push-pull model, as evidenced by the presence of tuning in the F0 components of the membrane potential of layer 4 neurons, and which accounts for the small residual broadening of orientation tuning curve at low contrast.

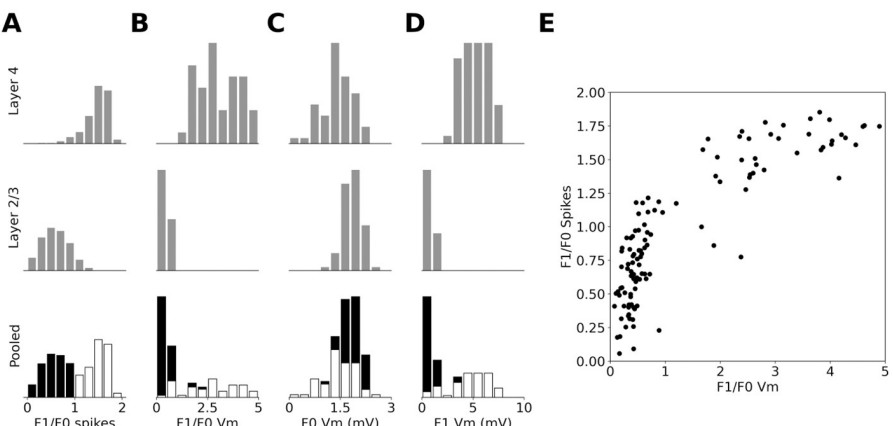

**Fig 8. MRs of excitatory cells in the model.** (A) Histogram of MRs calculated from the PSTH over recorded excitatory cells in Layer 4 (first row), Layer 2/3 (second row) and pooled across both layers. (B) as in (A) but MRs calculated from the membrane potential after subtraction of background level. (C) As in A but the histograms are of the absolute value of the F0 component of the membrane potential. (D) As in C but for the F1 component of the membrane potential. (E) The relationship between the MR calculated from PSTH and from membrane potential. In the pooled histogram plots cells classified as Simple are marked white and as Complex are marked black.

**2.2.3 Simple and Complex cell types.** Next we examined the classification of the cells into Simple and Complex. The modulation ratio (MR) of the peri-stimulus time histogram (PSTH) classified most model Layer 4 neurons as Simple cells (see Fig 8A) and most Layer 2/3 neurons as Complex cells (see Fig 8A), in-line with the experimentally observed laminar biases in cat [53]. When pooled across the two layers the histogram of MRs forms the characteristic experimentally observed bimodal distribution (see Fig 8A; [54]).

Unlike for the PSTH, the MRs computed from the membrane potential (with the resting potential subtracted) and pooled across the cortical layers form a unimodal distribution (see Fig 8B), in line with experimental evidence in cat [54]. When the MRs calculated from PSTH and membrane potential are plotted against each other (Fig 8E), a characteristic hyperbolic relationship is revealed, in line with the observations of Priebe *et al.* [54].

We also analyzed the F0 and F1 components of the membrane potential individually. Both have a unimodal distribution as observed experimentally by Priebe *et al.* [54]. The range of values for the F1 component fits the experimental data well, as Priebe *et al.* [54] observes that most complex cells have an F1 component lower than 5 mV and that only few simple cells have an F1 component higher than 8 mV. On the other hand, the distribution of the F0 component of membrane potential (Fig 8C and 8D) is narrower in both modeled layers than observed experimentally [54]. This could be due to intrinsic regularities present in the model, such as identical physiological parameters among all cells of the same cell type, or limited variability of the ratio of afferent and recurrent inputs to model Layer 4 neurons, or completely phase-non-specific connectivity from Layer 4 to Layer 2/3. These artificial inherent regularities of the model could be exaggerating the differences between the two functional neural types.

## 2.3 Size tuning

Next we studied the response of model neurons to DGs confined to an aperture of increasing diameter, to assess the presence of surround suppression in the model. While most excitatory neurons in both cortical layers show surround suppression (Fig 9A–9D), we observe a diversity of tuning patterns. An example of a Layer 4 cell with prototypical size tuning properties,

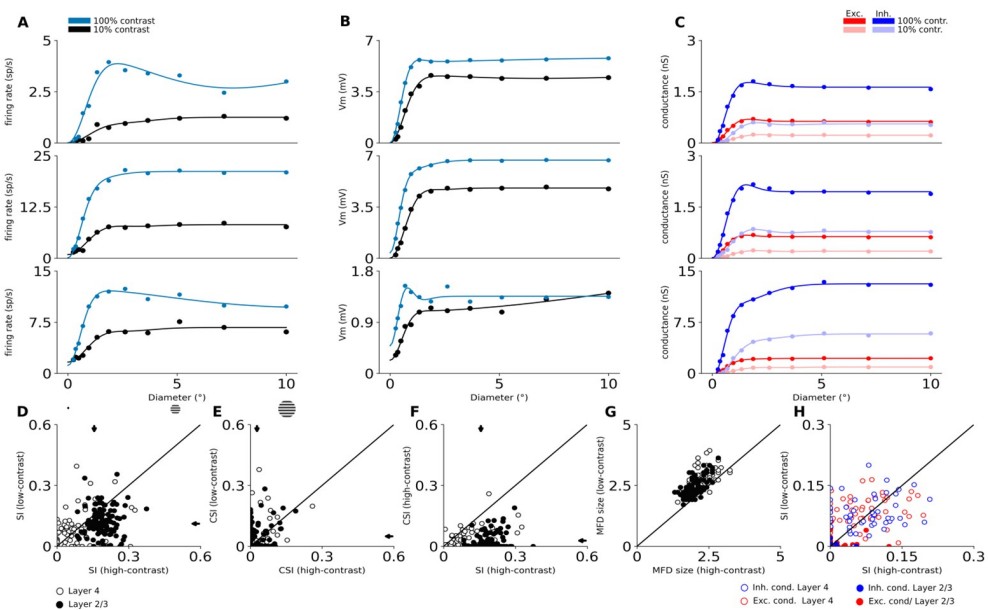

**Fig 9. Size tuning in excitatory neurons.** The first three rows show size tuning in three example model neurons. Top row is a typical Layer 4 cell with strong size tuning, second row shows an example of Layer 4 cells that does not exhibit size tuning, and the third row shows a typical Layer 2/3 cell. (A) The size tuning curve of trial-averaged spiking responses. Dots are measured responses, line is fitted size tuning curve. (B) The size tuning curve of trial-averaged mean membrane potential. (C) The size tuning curves of trial-averaged mean excitatory (red) and inhibitory (blue) synaptic conductances. Data shown for low and high contrast conditions: (A-B) low-contrast (10%) condition is black, high-contrast (100%) condition is blue; (C) low-contrast (10%) condition is un-saturated, high-contrast (100%) condition is saturated; (D-H) Scatterplots of size-tuning measures. Arrows mark means after pooling across layers. White (empty in H) dots correspond to individual Layer 4 neurons and black (full in H) dots correspond to Layer 2/3 neurons. (D) The suppression index of spike responses at low vs. high contrast. (E) The counter-suppression index of spike responses at low vs. high contrast. (F) The suppression index vs. the counter-suppression index at high contrast. (G) the maximum facilitation diameter at low vs. high contrast. (H) The suppression index of excitatory (red) and inhibitory (blue) synaptic conductances at low vs. high contrast.

exhibiting the expansion of facilitation diameter at low contrast [55, 56], is shown in the first row of Fig 9, whereas the second row of Fig 9 shows an example of a Layer 4 cell that exhibits only weak suppression. Neurons in Layer 2/3 also express classical size tuning effects, as exemplified in the third row of Fig 9.

The degree of the suppression varies widely in both cortical layers (Fig 9D). Overall the suppression is stronger in Layer 2/3 (mean suppression index (SI) at high contrast: 0.19 ± 0.004) than in Layer 4 (mean SI at high contrast: 0.10 ± 0.010). These values are on the lower end of the range of experimental observations in cat V1 (i.e. 0.16 [56], 0.35 [57], 0.44 [55] and 0.47 [58]; cells pooled across all cortical layers). This could come from the absence of higher visual areas in our model, which feedback connections have been shown to contribute positively to surround suppression in macaque V1 [59]. Stronger suppression in Layer 2/3 than in Layer 4 was observed in three studies [57, 58, 60] while [55] did not observe a statistically significant difference between the layers. 94.1% of the cells of the model are suppressed for large stimuli as they display SI greater than 0, in great quantitative agreement with the value of 95% observed experimentally [55]. Finally, Wang *et al.* [55] observed a decrease of suppression at low contrast, which is consistent with the results of the Layer 2/3 of our model (mean SI at low contrast in Layer 2/3: 0.12 ± 0.006; mean SI at low contrast in Layer 4: 0.10 ± 0.011).

At high contrast, we observe that the mean maximum facilitation diameters (MFD) for Layer 4 and Layer 2/3 neurons are respectively 2.30˚ ± 0.06˚ and 2.00˚ ± 0.03˚ (Fig 9G). It is

difficult to precisely compare these numbers to experimental data, as RF size in cat V1, and hence MFD, grows with neurons' foveal eccentricity [61], but we are not aware of any study that reports eccentricity-dependent data on size tuning. Studies have reported a broad range of mean MFD values (2.9˚ [62], 3.5˚ [60], 6˚ [56]). The pooled mean MFD we observe in our study is on the lower end of this range, which is consistent with the fact that all these studies recorded neurons at (on average) larger eccentricities than the 3˚—at which our model is centered (see Methodology). We observe important trends in the model that are consistent with experimental data. First, we observe an increase of the MFD at lower contrast (Fig 9G; mean MFD at low contrast: 2.84˚ ± 0.06 Layer 4 and 2.63˚ ± 0.04 Layer 2/3). This change of 27% (pooled across layers) is in good agreement with Wang *et al.* [55] and Tailby *et al.* [56] who found increases of 33% and 36% respectively in cat V1. Second, we observe smaller MFD in Layer 2/3 than in Layer 4 in line with experimental observations [60]. Finally, we find that both excitatory and inhibitory conductances are size tuned in both simulated cortical layers (Fig 9H), in line with experimental evidence [63, 64].

Many neurons do not exhibit monotonic suppression after reaching their classical RF size: after a certain diameter their responses partially recover from the suppression [55]. This phenomenon has been named counter-suppression. We observe this in a substantial subset of model neurons across both modeled layers (Fig 9E), as shown also in some of the single neurons examples (first row in Fig 9A). When quantifying the magnitude of this counter-suppression as the CSI index, we find that it is stronger at low contrast (mean CSI at low contrast: 0.05 ± 0.009 Layer 4 and 0.05 ± 0.005 Layer 2/3) than at high contrast (mean CSI at low contrast: 0.03 ± 0.006 Layer 4 and 0.02 ± 0.004 Layer 2/3), which is in line with Wang *et al.* [55]. It should be noted, however, that the limited area of the simulated cortical network likely prevents the full build-up of counter-suppression effects.

## 2.4 Trial-to-trial variability: Response to natural images with simulated eye-movements

Next we probed the model with a stimulus consisting of a natural scene animated with simulated eye-movements, as described in the intra-cellular study of Baudot *et al.* [21]. As can be seen in Fig 10B, the natural image stimulus elicits a highly repeatable response in cat V1, both at the level of spikes and at the level of sub-threshold responses. In contrast, the response to presentation of a DG of medium contrast (30%) is only locked to the slow temporal frequency of the luminance modulation (Fig 10A). This stimulus dependent difference in response statistics is qualitatively consistent with the findings of Baudot *et al.* [21]. The presence of a strong short-term synaptic depression in the thalamo-cortical connections shapes the response in a stimulus-dependent way, as it is less triggered by the transient changes in the natural scene stimulus than by the sustained activation induced by the DG [65].

To further investigate the response precision and reliability of the model neurons we computed the cross-correlation between trials of the spiking and membrane potential responses. The reliability is given by the peak amplitude of the cross-correlation ($a$) at time zero, and the temporal precision by the standard deviation of the Gaussian fit ($\sigma$) [21]. As shown in Fig 10E and 10F, for the spiking response (top), both for model Layer 4 and Layer 2/3 neurons, the cross-correlation has a higher peak and is narrower for the natural-image-with-eye-movement (NI) stimulus (Layer 4: $a$ = 0.221, $\sigma$ = 12; Layer 2/3: $a$ = 0.081, $\sigma$ = 15), than for DGs at medium contrast (Layer 4: $a$ = 0.187, $\sigma$ = 60; Layer 2/3: $a$ = 0.037, $\sigma$ = 60), in line with the experimental results [21] (see Fig 10H).

For the membrane potential (Fig 10E and 10F bottom) the same relationship holds except that for Layer 4 neurons the peak (and thus the reliability) for gratings is higher than for

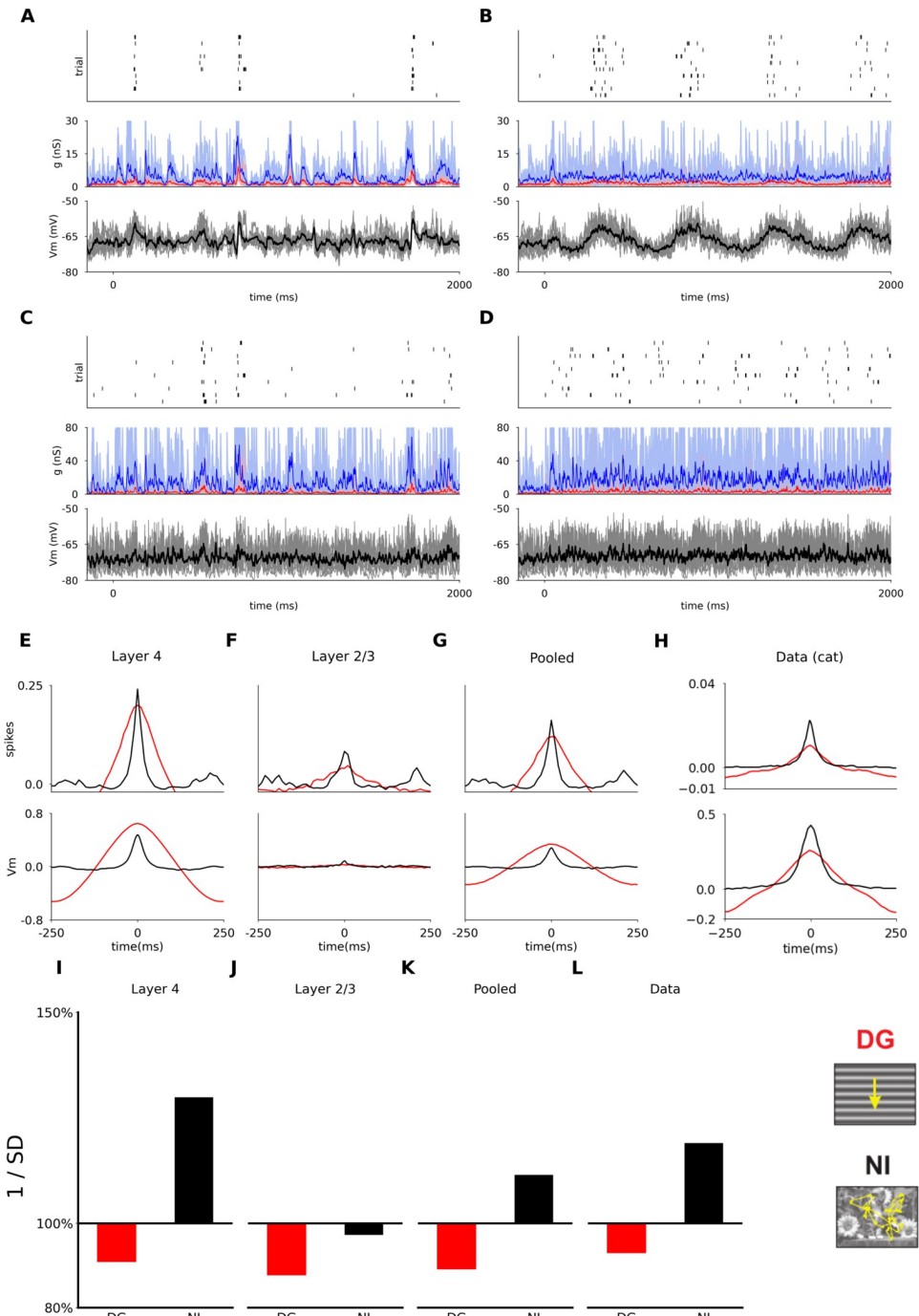

**Fig 10. Trial to trial reliability of responses to a natural image with simulated eye movements (NI) and to drifting sinusoidal gratings (DG).** (A) The response of a randomly sampled excitatory neuron in model Layer 4 to the NI stimulus. The top panel shows the spike raster histogram, the middle panel shows the incoming synaptic excitatory (red) and inhibitory (blue) conductances. The bottom panel shows the membrane potential. The thick lines show the mean of the analog signals over the 10 trials. (B) As (A), but for the DG stimulus (contrast 30%). (C-D) As (A-B) for an excitatory neuron in Layer 2/3. (E-H) Cross-correlation across trials for spikes (top) and subthreshold membrane potential activity (bottom), in response to DG and NI stimuli. The reliability is given by the peak amplitude at time zero, and the temporal precision by the standard deviation of the Gaussian fit to the cross-correlation. (E) Results averaged across all recorded excitatory neurons in model Layer 4. (F) Results averaged across all recorded excitatory neurons in model Layer 2/3. (G) Results averaged across all recorded excitatory neurons pooled across the two model layers. (H) The experimental data from [21]. Note that in that study two additional stimuli were also presented (dense

noise: blue line; grating with simulated eye-movements: green line) that we have not used in this study. (I-L) The inverse of the stimulus-locked time-averaged standard deviation of membrane potential (1/SD) across trials, averaged across cells, relative to the value for ongoing activity. (I) Results averaged across all recorded excitatory neurons in model Layer 4. (J) Results averaged across all recorded excitatory neurons in model Layer 2/3. (K) Results averaged across all recorded excitatory neurons pooled across the two model layers. (L) Experimental results in V1 of the anesthetized cat [21].

natural images (Layer 4 NI: $a$ = 0.441, $\sigma$ = 18; Layer 2/3 NI: $a$ = 0.064, $\sigma$ = 15; Layer 4 DG: $a$ = 0.63, $\sigma$ = 99; Layer 2/3 DG: $a$ = 0.025, $\sigma$ = 71), unlike the observations in [21] (Fig 10H), where the reliability is higher for the NI stimulus for both spikes and membrane potential.

Next, Baudot *et al.* [21] found that relative to the trial-to-trial variability of membrane potential during spontaneous activity, the variability of membrane potential during DG stimulation increases, while it decreases during NI stimulation (Fig 10L). Baudot *et al.* [21] quantified the variability modulation as the inverse of the stimulus-locked time-averaged standard deviation across trials (1/SD), and expressed as a percentage relative to the value for ongoing activity, meaning that values below 100% reflects an increase of trial-to-trial variability. The results of the same analysis in the model are shown in Fig 10I and 10J. In both model layers we observe an increase of stimulus-locked variability for DG (relative 1/SD equal to 91 ± 1.05% in Layer 4 and 88 ± 0.2% in Layer 2/3). For the NI condition, we observe a decrease of stimulus-locked variability in Layer 4, as well as a small increase in Layer 2/3 (relative 1/SD equal to 130 ± 0.53% in Layer 4 and 97 ± 0.22% in Layer 2/3). However, note that in Baudot *et al.* [21] the results are pooled across neurons of all layers, and the same treatment of our simulation data yields a good quantitative match to the experimental data (Fig 10K).

The lack of direct external noise injected into cortical model neurons was crucial for achieving the results presented in this section (panel C in S2 Fig). The trial-to-trial variability in the model is primarily shaped by the intra-cortical interactions during both spontaneous and stimulus driven activity. This is an advance over a previous model of a V1 column that explained the precision and reliability changes between the grating and naturalistic stimuli at the spiking level, but did not reproduce the stimulus-dependent variability effect at the level of membrane potential [65]. This was due to the inclusion of substantial random spiking input directly into cortical neurons, which induced substantial stimulus-independent variability of the membrane potential, overriding the internally generated variability components. These differences point out the importance of simultaneously capturing within a single model the cortical processing under multiple stimulation conditions and at different levels of signal integration, to be able to fully describe its operation.

## 3 Discussion

A coherent explanation for how the numerous V1 computations coexist within the underlying neural substrate is still lacking. We have approached this problem with a systematic, integrative, data-driven methodology, constructing a model of V1 firmly grounded in anatomical and physiological data. We show that a single set of plausible connectivity principles—(i) weakly orientation-biased thalamic input, (ii) local, weakly push-pull-biased intra-Layer 4 connections, and (iii) long-range weakly orientation-specific intra-Layer 2/3 connections—give rise to the most salient V1 computations. This single model instance provides insight into the underlying mechanisms of processing within the classical receptive field: orientation tuning of both spiking and subthreshold signals (Figs 6 and 7); and emergence of Simple and Complex cells (Fig 8). The same model gives an accurate account of computations requiring integration beyond the classical receptive field: size tuning of both spikes and synaptic conductances (Fig

9). Finally, beyond artificial stimuli, the model replicates the stimulus-dependent changes to the precision of spiking responses, and to the trial-to-trial variability of membrane potential (Fig 10) when stimulated by a naturalistic spatio-temporal stimulus. All these stimulus evoked properties are underpinned by a resting state that is in very good quantitative agreement with physiological data at multiple levels of integration (Fig 4).

## 3.1 Integrative computational modelling approach

To support model component reuse, and hence the long-term integrative modeling program proposed in this study, we built our model in the Mozaik neural network modeling environment [17, version 0.4], optimized for efficient specification and reuse of model components, experimental and stimulation protocols and model analyses, and are making the resulting model code available under a liberal open-source license. We also make available our library of integration tests via a dedicated data store (http://v1model.arkheia.org) implemented in the recent Arkheia framework [66], sharing model and virtual experiment specifications. The integration tests are structured so as to be independent of the model specification, and their ready-to-use implementation in the Mozaik environment is provided. Altogether, we provide here an advanced and ready-to-use platform for future long-term, systematic, incremental, integrative research on the early visual system of higher mammals and beyond.

## 3.2 Novelty and predictions

The central contribution of this study is the original convergence of the following key integrative principles: (i) the multitude of anatomical constraints respected, (ii) the multiplicity of neural signals (synaptic, single unit) and integration levels (conductance, neuronal, columns) validated, (iii) the multiplicity of the stimulation protocols tested, (iv) the diversity of functional phenomena explained, and (v) the multiplicity of spatial and temporal scales validated. While many of the properties demonstrated in this study have been previously modeled in isolation, to the best of our knowledge, the single V1 model instance presented here was validated with the broadest range of visual stimulation protocols to date (Table 1). As such, the present study represents a significant advance towards providing a comprehensive integrative treatment of the early visual system.

This model also provides a description of the asynchronous irregular resting regime that is quantitatively accurate at both supra- and sub-threshold levels, does not rely on *ad-hoc* noise input to cortex, and is compatible with evoked processing. This allows us to demonstrate, how the trial-to-trial variability of membrane potential in V1 neurons can increase during drifting grating stimulation and decrease during naturalistic movie stimulation [21].

This model also provides a mechanistic explanation of orientation tuning that is simultaneously consistent with all the following physiological constraints: (i) orientation tuning is sharp across cortical layers [49], (ii) inhibition is only moderately broader than excitation [49, 79], and (iii) both F0 and F1 components of spikes, membrane potential and conductances tend to be tuned to the cell's preferred orientation. Many past models have only explored the orientation tuning of spiking responses [80, 81] and relied on either broad inhibition [5, 50, 82] or inhibition tuned to the orthogonal orientation [83, 84], which is in contradiction with inhibitory conductances tuned to preferred orientation [51]. Few past models have been consistent with the constraint of sharply tuned responses of inhibitory cells and of inhibitory input tuned to the preferred orientation [85], and no modelling studies demonstrated the orientation tuning of both the F0 and F1 components of all the sub-threshold signals.

The model offers a number of testable predictions about properties of cat V1:

**Table 1. Validation of visual cortex models with respect to experimentally observed functional properties.** A green tick means that the model gives a qualitatively good match to experimental findings. A red cross means that the model gives a response which is qualitatively different from experimental findings. A red open circle means that the model does not incorporate neural structures necessary for this feature to be meaningfully tested. Empty spaces mean that the model was not tested for the property, and it is not possible with certainty to conclude from the corresponding paper that the model does or does not match the corresponding experimental data. When the species is clear, a dagger next to the model name means that the model was based on cat V1, whereas double daggers mark models based on primate V1.

| | This study† | Troyer 1998† | Somers 1998† | Wielaard series [67, 68]‡ | Tononi series [69–72]† | Rangan series [13, 73–75] | Kremkow 2016† | Chariker series [50, 76–78]‡ |
|---|---|---|---|---|---|---|---|---|
| Spontaneous state without ad-hoc noise source | ✓ | ✗ | ✗ | ✗ | ✓ | ✗ | ✗ | ✓ |
| Realistic levels of cond. in the spontaneous state | ✓ | | | | | | ✓ | |
| Log-normally distributed spont. firing rates in L2/3 | ✓ | ○ | ○ | ○ | | ○ | ○ | ○ |
| Realistic mean rate in spontaneous state in L4 | ✓ | | | | ✓ | | | ✓ |
| Realistic mean rate in spontaneous state in L2/3 | ✓ | ○ | ○ | ○ | ✓ | ○ | ○ | ○ |
| Spontaneous patterns of cortical activity | | | | ✓ | ✓ | | | |
| Bimodal distribution of MR (spikes) | ✓ | ✗ | | ✓ | ✗ | ✓ | ✗ | ✓ |
| Unimodal, monotonic distribution of MR (Vm) | ✓ | | | ✓ | | ✓ | | |
| Layer 4 orientation tuning width | ✓ | ✓ | | | ○ | ✓ | ✓ | ✓ |
| Layer 2/3 orientation tuning width | ✓ | ○ | ○ | ○ | ○ | ○ | ○ | ○ |
| Contrast invariant orientation tuning | ✓ | ✓ | | | ○ | ✓ | ✓ | |
| Conductance increase at onset & offset of flashed bar | | | | ✓ | | | | |
| Anti-phase cond. in Simple cells responding to DG | ✓ | ✓ | ✗ | ✗ | ✗ | | ✓ | ✗ |
| Spatial frequency selectivity | | | | ✓ | | | | ✓ |
| Direction selectivity (DS) | | | | | | | | ✓ |
| DS broadband across spatial frequencies | | | | | | | | ✓ |
| DS broadband across temporal frequencies | | | | | | | | ✓ |
| Precision and reliability of spike responses to NI | ✓ | | | | | | ✓ | |
| Stimulus dependent trial-to-trial variance of Vm | ✓ | | | | | | ✗ | |
| Gamma band synchronization | | | | | ✓ | | | ✓ |
| Size tuning | ✓ | ○ | ✓ | ✓ | | | ○ | |
| Contrast dependent surround-suppression | ✓ | ○ | ✓ | ✓ | | | ○ | |
| Contrast dependent shift in size tuning peak | ✓ | ○ | ✓ | ✓ | | | ○ | |
| Size-tuning of exc. and inh. conductances | ✓ | ○ | | ✓ | | | ○ | |
| Size-tuning counter-suppression | ✓ | ○ | | | | | ○ | |
| Line-Motion illusion | | ○ | | | | ✓ | ○ | |
| Slow Wave Sleep mode with up & down states | | | | | ✓ | | | |

*(Continued)*

**Table 1.** (Continued)

| | This study† | Troyer 1998† | Somers 1998† | Wielaard series [67, 68]‡ | Tononi series [69–72]† | Rangan series [13, 73–75] | Kremkow 2016† | Chariker series [50, 76–78]‡ |
|---|---|---|---|---|---|---|---|---|
| Bi-modal distribution of Vm in sleep mode | | | | | ✓ | | | |
| Slow oscillation propagation in cortical space | | | | | ✓ | | | |

1. Layer- and cell-type-specific differences in resting regime:

   (a). Higher inhibitory conductances in Layer 2/3 than in Layer 4.

   (b). Greater synchrony among the inhibitory neurons.

2. Layer-specific distributions of MRs of spiking and membrane potential responses (layer specific data is missing in cat).

3. Near contrast-invariance of inhibitory neurons.

4. Layer-specific differences in sub-threshold signal orientation tuning.

5. Layer-specific differences in conductance size tuning.

6. Important testable assumptions were made about the model parameters:

   (a). The sharpness of functional specificity of cortico-cortical connections.

   (b). The excitatory-to-excitatory vs. excitatory-to-inhibitory pathway strength ratio.

## 3.3 Other integrative models of V1

In the following, we present a cross-comparison of the presented model with the most prominent integrative models of V1 to date. We restrict this comparison to models of cat and macaque V1, as the two species represent the two major animal models for vision in higher mammals, while their anatomical and functional similarity allows for straightforward comparison.

Somers *et al.* [12] investigated a topologically-organized model of V1, which incorporated excitatory connections that dominated at very short and long distances, and inhibitory ones that dominated at intermediate distances. Direct anatomical evidence for such an arrangement of lateral connections is lacking [20]. All connection probabilities were biased towards co-oriented targets, but were independent of the phase preference of targets, which implies that the model was not able to replicate the anti-phase relationship between excitatory and inhibitory conductances [44, 51, 86]. The Somers *et al.* [12] model was able to demonstrate the emergence of contrast-invariant orientation tuning, and contrast dependence of size tuning, but none of the other features of V1 processing considered here were examined.

Another similar pair of modeling studies was performed by Wielaard *et al.* [67, 68]. These modeling studies did not consider long-range cortical interactions, nor functional specificity of intra-cortical interactions, implying that the models were not able to replicate the anti-phase of excitatory and inhibitory conductances [44, 51, 86]. Wielaard *et al.* [67] demonstrate the emergence of both Simple and Complex cell types and show how a bimodal distribution of MRs at the level of spiking emerges despite unimodally distributed MRs of membrane potential. Furthermore, they show how such an only-locally-connected cortical model can explain

the emergence of size tuning, quantitatively well matching experimental results. Interestingly, the authors do not demonstrate some of the more classical properties of V1 neurons, such as sharp mean orientation tuning (the limited data presented indicate a predominance of poorly tuned cells) or its contrast invariance. Similar to our model, Wielaard et al. [68] demonstrate that the size tuning arises as a complex dynamic interplay between excitation and inhibition, whereby both conductances exhibit reduction at large stimulus sizes. Unfortunately Wielaard et al. [67, 68] do not assess the sub-threshold signals quantitatively, or characterize the background activity in the model. Finally, no naturalistic stimuli were tested in the model.

The Tononi group has also published a series of models focusing on the cat early visual system [69–72]. In addition to the LGN and cortical layers 4 and 2/3 considered in our model, the Tononi et al. [69–72] model series also incorporated a model of sub-granular layers and a rudimentary model of a generic extra-striate area that represented the totality of feedback input from higher-level visual areas to V1. Tononi et al. [69–72] aimed primarily at simulating the spontaneous activity during awake vs. slow wave sleep conditions. Accordingly, the model incorporated only rudimentary functionally specific circuitry, chiefly restricted to the classic Hubel & Wiesel type convergence of thalamic afferents onto cells in granular and infragranular layer that induced the most basic form of orientation selectivity. Only two orthogonal orientations assigned to two spatially separated cell groups were considered, and no other investigation of evoked activity was performed. On the other hand, the model demonstrated an impressive range of activity dynamics present in awake and slow wave sleep brain states including spontaneous patterns of cortical activity, slow wave sleep mode with up & down states, the slow oscillation propagation in cortical space, bi-modal distribution of membrane potential in sleep mode and gamma band synchronization in both sleep and awake modes.

One of the most comprehensive V1 modeling effort prior to the present study is a series of modeling studies [73–75] summarized in Rangan et al. [13], architecturally very similar to the Wielaard et al. models [67, 68]. While the models also focus on spontaneous regime dynamics, they rely on constant independent noisy spike input in the cortical neurons. In multiple related models the authors have demonstrated a range of V1 properties in both spontaneous and evoked states including a fluctuation-driven spontaneous activity regime characterized by low firing rates [73], emergence of sharply tuned Simple and Complex cell types with a bimodally distributed MR of spike responses and unimodally distributed MR of membrane potential [74], and approximate contrast invariance of orientation tuning. The study also demonstrated two features of V1 computation not examined in our model: the spontaneous activity dynamics that are correlated with the intrinsic orientation organization of cortex [73], and the spatio-temporal patterns of cortical activity in response to a line-motion illusion stimulus [75]. Unlike in the present study, neither the contextual modulation nor naturalistic stimulation were explored in these models. Most importantly, a detailed quantitative account of their sub-threshold behavior was not provided. Finally, while all the models summarized in Rangan et al. [13] share a common architecture, it is unclear whether all these studies used identical model parameterizations, and hence if all these features could be achieved within a single model instance.

Finally, recently Chariker et al. [50, 76–78] have also made the case for a more comprehensive approach to the computational study of V1, presenting a model that is, similarly to the present study, firmly grounded in anatomical data. Their work is complementary to ours as they model layer $4c\alpha$ of monkey V1, which receives sparse input from magnocellular cells. Their model displays both gamma oscillations [76] and realistic direction selectivity [78], which we did not investigate in our model. The authors do not probe the model with any stimulus other than the ubiquitous fullfield DG, and therefore do not investigate contextual modulation effects nor the response of their model to natural images. The characterization of the spontaneous state is missing. The model Simple cells also do not exhibit the anti-phase

relationship between excitatory and inhibitory conductances. The authors do investigate contrast dependence of orientation tuning, but only by examining the ratio of the orthogonal response to the preferred response, a measure to which V1 neurons are naturally not contrast invariant to [77]. It is therefore unclear whether the neurons in their model display contrast-invariance of the HWHH measure instead. Finally, the analysis of the model properties at the sub-threshold level is limited.

### 3.4 Other models of V1

Models of orientation selectivity are typically categorized according to their circuitry ('feed-forward' vs 'recurrent'), the relative impact of such feed-forward vs recurrent contributions on the genesis of orientation selectivity, and the selectivity of the cortical inhibition. In 'feed-forward' models [5, 80, 87] orientation tuning is induced by a strong sharply tuned thalamo-cortical pathway, while intra-cortical connections have only a minor influence. In contrast, 'recurrent' models [88, 89] assume weak poorly tuned thalamic input but strong intra-cortical interactions. With respect to inhibition, the defining features are its orientation tuning, its presence or absence at orthogonal orientations, and its relationship to stimulus phase.

Although the opposition between thalamic-driven and cortical-recurrent computing schemes has long dominated the literature, this dichotomic categorization is highly oversimplified [7]. Following the anatomical and physiological evidence to determine the model architecture, as we have done here, results in a model that resides in between these standard classes. The thalamo-cortical connections in the model constitute a fraction of the synaptic input to Layer 4 ($\sim 10\%$), which would assign the model to the recurrent category. On the other hand, the dominating intra-cortical connections in Layer 4 are biased towards a push-pull organization, typically assumed in feed-forward models [5] (although, based on experimental evidence [18, 90], this functional bias is weaker than in previous models). Consequently, the inhibitory input to Layer 4 neurons is broad and present at orthogonal orientations. But the resulting orientation tuning of inhibitory neurons is sharp, comparable to that of excitatory ones, showing that even weak and poorly orientation-tuned afferent input and broad inhibitory input can lead to sharply tuned inhibitory neurons, in line with experiments [49, 51].

Furthermore, in classical feed-forward models the F0 component of the feed-forward excitatory input is canceled-out by the F0 component of the inhibition [5]. It is the F1 component of the membrane potential that induces the orientation tuning. But, in line with experimental evidence, our model still exhibits orientation tuning of the F0 components of both membrane potential and excitatory and inhibitory conductances, suggestive of a boosting of the mean response at preferred orientations, typical of recurrent models. However, the push-pull interactions remain an important factor in the generation of contrast-invariant orientation tuning, as evidenced by the dominance of the F1 components in Simple cells. This shows that different strategies of generating orientation tuning can co-exist within the same neural circuit, facilitating the models' consistency with a broader range of physiological findings, hence exemplifying the benefits of the data-driven integrative approach in counteracting the fragmentation of knowledge.

Recently, a feed-forward explanation of contrast-invariant orientation tuning was proposed by Finn *et al.* [8], relying on two key mechanisms: an expansive non-linearity governing the relationship between membrane potential and the spike rate, and the contrast dependence of the thalamic input. However, although the formulation of the Finn *et al.* [8] model is feed-forward, the assumed non-linearities may in the cortical substrate be implemented by recurrent dynamics. Furthermore, the Finn *et al.* [8] model does not explain the fact that the modulation of the membrane potential is dominated by inhibition, nor the anti-phase relationship between inhibition and excitation [44, 51, 86].

Finally, recent modeling studies from the Allen Institute [91, 92] also followed a comprehensive, data-driven approach, employed a broad set of test protocols, and demonstrated orientation selectivity, amplification of thalamic inputs, presence of gamma oscillations, and log-normal distributions of firing rates. Unlike the present study they did not investigate more complex functional properties, such as size-tuning or trial-to-trial precision. Their models are complementary to the model presented here: cat V1 in the anesthetized state vs awake mice [91] (respective merits of the two preparations are discussed in section 3.5). While Arkhipov *et al.* [91] model was restricted only to Layer 4, Billeh *et al.* [92] modeled all V1 layers. Both studies also incorporated neural morphology, although they showed that a simplified, integrate-and-fire version of their model remained in good agreement with *in-vivo* data, justifying the level of detail chosen in the present study.

## 3.5 Modeling choices

We have chosen to focus our model on the cat visual system, but supplement the model with data from macaque whenever equivalent data from the cat is not available, as the closest available approximation. The most extensive body of data on V1 has been collected in cat, including several of the key datasets for this study, such as extensive parametric studies of visual RFs, the extents of intra-areal connectivity in V1 [20] and the relative strength of the different intra-areal pathways [19]. Furthermore, there is greater availability of in-vivo intracellular data, which are critical for constraining our model, from the anesthetized cat [44, 51, 86, 93–95]. Finally, cat area 17 can be equated with macaque area V1, based on criteria of relative position in the cortical mantle, internal organization of visual field representations, and trans- and sub-cortical connections.

The level of biological detail of the presented model implies a substantial set of model parameters that have to be determined. The process of parameter determination was divided into two stages. In stage one we first determined the parameter values or ranges for all model parameters from the experimental literature. Through this process we obtained constraints for an absolute majority of model parameters including: (i) physiological parameters of the neural model which were primarily based on intracellular patch clamp recordings in [44] (Section 4.4.1), (ii) spatial distributions of local connectivity between different cortical layers and neural types [96] (Section 4.1.4), (iii) functional specificity of layer 2/3 connectivity, based on [18] (Section 4.1.5), and (iv) synaptic delays, based on a set of paired recording studies (Section 4.1.7).

Unfortunately, current experimental evidence is insufficient for precise determination of every single model parameter. In this study we hypothesize the existence of push-pull connectivity among Layer 4 neurons, as it is the only plausible explanation for the anti-phase relationship between excitatory and inhibitory conductances in Simple cells [21, 97] proposed to date (Section 4.1.5). As this circuitry has not been directly probed, its hypothetical parameters are unknown. Another poorly constrained parameter is the unitary synaptic strength between neurons. Existing experiments have reported a wide range of values, only indicative of the order-of-magnitude of the synaptic strength, and a systematic identification of the synaptic strength with respect to pre- and post-synaptic neuron type and layer membership is completely missing (Section 4.1.6).

In stage two we fine-tuned the parameters that were not fully determined in stage one. The tuning was guided primarily by the goal of achieving stable dynamics and physiologically plausible mean values of intra and extra cellular signals in the spontaneous condition. In our experience, achieving physiological behavior in the spontaneous state secured physiological behavior also in the evoked conditions. While the model dynamics were robust to many of the parameters, we noticed that the model was particularly sensitive to parameters that governed the balance between excitation and inhibition, such as the synaptic strengths between different neural types. However, the broad range of stimulation protocols and functional features that

the models is tested against ensures that this fitting process avoids major over-fitting that could skew the interpretation of the results.

Having instantiated the model based on the parameterization determined in the process described above, we next validated the model against a broad set of experimental findings. Acknowledging the finite manpower resources, and the fact that the integrative program is inherently an incremental process, we identified a priority list of V1 phenomena to be addressed in this first snapshot. These validation tests were strategically selected to cover the most common stimulation paradigms (the stimuli represent a superset of visual stimulation protocols used in our own lab [21] and in the Allen Institute's Brain Observatory [98]) and a broad range of the most salient and well established functional features of early visual system.

## 3.6 Model limitations

The model makes five main simplifications: (i) the omission of layers 5 and 6 [99, 100], (ii) omission of the cortico-thalamic feedback pathway [101, 102], (iii) omission of the feedback from higher cortical areas [103, 104], (iv) reduction of neural dynamics to only two types (excitatory and inhibitory) [79, 105], and (v) the reduced cortical neuron and synaptic densities. The infra-granular layers have been omitted because they are much less understood and there is much less data available for constraining their model. Furthermore, they are believed to be largely involved in sub-cortical projections and feedback projection from higher level cortical areas [106], which are another two features not considered here (but see [107]). The cortico-cortical feedback, mediated partly by long-distance supra-granular connections, represents a particularly difficult challenge for modeling, as it in principle requires some form of treatment of all the numerous higher-cortical areas that send feedback to V1, while relying on a relatively sparse set of findings on the anatomy and physiology of these feedback structures. Due to recent methodological advances in selective targeting of different neural subtypes, the roles which the different inhibitory neural sub-types play in cortical processing are starting to be uncovered [108, 109]. This mapping out of the neural sub-type characteristics is, however, still in an early stage, and is largely confined to the mouse, without a clear understanding of how these findings will translate to cortical processing in higher mammals. Undoubtedly, continuing systematic exploration of the neural sub-type specificities, especially if successfully translated into higher mammalian models, would provide an invaluable trove of constraints to be incorporated into the model.

The need to model a large enough V1 surface to cover sufficiently long-range lateral interactions and contextual modulation, while keeping the computational complexity of the model manageable (to allow the simulation of experiments lasting thousands of seconds of biological time), lead to important compromises. We have decided to model only roughly one-tenth of neurons present in the cortical layers covered by our model, while the 1000 synapses modeled for layer 4 neurons and 2300 for layer 2/3 neurons potentially under-estimate the true number of local connections even if transmission failures are factored in (see Materials and methods). In addition, connections coming from outside of V1 were not modeled at all. While the results presented in this study show that the lower neural density is not a limiting factor for the emergence of the range of phenomena addressed here, previous computational studies showed that the neural density can have an important impact on the correlation structure of the network activity [110], or impact the statistics of spontaneous cortical waves [111]. It is thus possible that, as we seek to explain an ever greater range of phenomena in the future, we will have to re-evaluate the neural density down-sampling.

The absence of a significant portion of the incoming connections in our model (primarily due to omission of extra-V1 innervation) raises several issues. To properly model all the

connections coming from extra-V1 areas would require us to model all those source areas, which is outside of the scope of this study. This however raises the question of how to compensate for the absence of the drive from the missing connections. A common solution is to model an extra independent Poisson spiking input into each cortical neuron [112], that compensates for the missing drive. This is quite reasonable when the rate of such compensatory input can be estimated as approximately Poissonian (e.g. spontaneous condition). However, for visually evoked conditions, where the magnitude of the external drive might depend on the origin of its source, the ongoing visual stimulus and likely even the specific function of the receiving V1 neuron, constructing a reasonable approximation of an extra-areal drive is extremely difficult. Furthermore, it is reasonable to expect that the visual stimulus heavily modulates the statistics of the stochastic extra-areal input, making an approximation with a stationary Poissonian input inappropriate. This latter issue is particularly problematic given that we wished to study the modulation of fine temporal statistics in the spontaneous vs. visual driven conditions.

To explore quantitatively the latter issue, we have repeated the experiment performed in section 2.4 that compares the trial-to-trial variability of the membrane potential in response to drifting sinusoidal grating vs. naturalistic stimulus (Fig 10), but this time in a model enriched with an external Poissonian input impinging on all cortical neurons at a rate of 1000 Hz (panel C in S2 Fig), which corresponds to a roughly 1 Hz spontaneous rate arriving via 1000 extra-areal connections. As can be seen, unlike in the original model (panel A in S2 Fig), the model with external input shows virtually no change of trial-to-trial variability for both the grating and naturalistic conditions in contradiction to the data (panel D in S2 Fig). This is because the constant, stimulus independent, external Poissonian noise drowns the stimulus dependent modulation of variability within the cortico-thalamic network.

For all these reasons our decision was to not include such compensatory external inputs in our model. Yet, as we show in section 2.1 and 2.2, our model reaches realistic levels of spontaneous and evoked activity at both the spiking, membrane potential and conductance levels. How is this possible? As we explain in section 4.1.6, the existing evidence on the synaptic strength of the various projections included in our model is very limited, leaving a lot of leeway for setting the exact synaptic weights. To deal with this unknown we set the synaptic weights to achieve an overall balance between excitation and inhibition that supports reasonable levels of both spontaneous and evoked activity, while being compatible with the limited physiological findings. It is thus possible that these broad constraints on synaptic weights allowed us to compensate for some of the external drive into stronger-than-biological synaptic weights. This points to a general limitation of reductionist modeling where missing aspects of the model are often (at least partially) absorbed into unrelated free parameters of the model.

We model neurons as exponential integrate-and-fire units with conductance-based synapses [113]. This choice had several motivations. As we have demonstrated here, a number of important characteristics of V1 computations are manifested at the precision level of single spikes, and such quantities do not have a direct representation in firing-rate based models. We also show here how the dynamics of sub-threshold signals, including the excitatory and inhibitory synaptic conductances, provide an important means for constraining the model, which thus excludes usage of current-based synapse models. The particular choice of the exponential variant of the conductance-based integrate-and-fire scheme was motivated by our observations that the variable effective threshold in this neural model secures more stable asynchronous behavior than the simpler fixed-threshold variants. We chose not to pursue more detailed representations of single neuron dynamics, such as a full morphological representation or ion channel kinetics, for two main reasons: (i) the neuron counts necessitated by the cortical area considered in this study, and the length of the stimulation paradigms targeted in

this study would make simulations with such detailed neural models nearly intractable on currently available computing resources, and (ii) lack of sufficient detailed reconstructions of neurons in cat V1 with associated intracellular functional data [114] and lack of detailed characterization of ion channel kinetics and ion channel dendritic distributions in cat.

Another limitation of our study is that we do not model the full spectrum of the RF properties present in V1. For example, we assume fixed size, spatial frequency and aspect ratio of the templates from which afferent connectivity (and thus 'afferent' RF) of Layer 4 V1 neurons is generated. We also assume that both the ON and OFF channels are of equal strength, even though recent electrophysiological investigations revealed the dominance of the OFF pathway and systematic variations of the ON-OFF pattern of RF following the orientation maps [115, 116]. All these structural parameters are known to have substantial biological variability [49, 117, 118]. They are the result of developmental processes, whether intrinsic or stimulus driven, which both have stochastic components, but more importantly likely generate features of cortical organization more broadly adapted to processing of visual stimuli that are not accounted for in our model (and presumably some or even most have not even been discovered yet). Unfortunately, quantitative estimates of these various sources of variability are not currently available. Our modeling paradigm could be used in the future to formulate predictions for these values by evaluating which additional sources of variability and of what magnitude could account for the experimental observations. An alternative intriguing possibility would be adoption of a hybrid scheme, whereby some of the variability due to the developmental processes would be directly simulated in dedicated adaptive models (e.g. see [10, 119]), and then transplanted into the detailed spiking modeling paradigm for more thorough examination.

Ultimately, the main reason for omission of these important structural features was to restrict this initial study to a manageable (both computationally and conceptually) level of complexity. We would like to emphasize that even though we argue here for an integrative, comprehensive treatment of the neural systems under study, we acknowledge that this has to be an incremental process. All these omissions would be natural candidates for the next steps of the integrative research program initiated in this study.

## 4 Materials and methods

The model's overall architecture was inspired by a previous rate-based model of V1 development [10] and a spiking model of a single cortical column [65]. The model is implemented, and all experiments and analysis defined in the Mozaik framework [17, version 0.4], which is freely accessible at https://github.com/CSNG-MFF/mozaik. The model can be run by first installing Mozaik and its dependencies as indicated in the Mozaik Github repository, and then by downloading the implementation of model at https://github.com/CSNG-MFF/mozaik-models/ [version 1.0] and following the instructions to run each experiment. The NEST simulator [120, version 3.4] was used as the underlying simulation engine.

### 4.1 V1 model

The cortical model corresponds to layers 4 and 2/3 of a $5.0 \times 5.0$ mm patch of cat primary visual cortex, and thus given the magnification factor of 1 at 3 degrees of visual field eccentricity [121], covers roughly $5.0 \times 5.0$ degrees of visual field. It contains 108150 neurons and $\sim 155$ million synapses. This represents a significant down-sampling ($\sim 10\%$) of the actual density of neurons present in the corresponding portion of cat cortex [122] and was chosen to make the simulations computationally feasible. The neurons are distributed in equal quantities between the simulated Layer 4 and Layer 2/3, which is consistent with anatomical findings showing that in cat primary visual cortex approximately the same number of neurons have cell bodies in

these two cortical layers [122]. Each simulated cortical layer contains one population of excitatory neurons and one population of inhibitory neurons in the ratio 4:1 [123, 124].

We model both the feed-forward and recurrent V1 pathways; however, overall the model architecture is dominated by the intra-cortical connectivity, while thalamocortical synapses constitute about 14% of the synaptic input to Layer 4 cells (Section 4.1.3), in line with experimental evidence [29]. The thalamic input reaches both excitatory and inhibitory neurons in Layer 4 (see Fig 1E and 1F). In both cortical layers we implement short-range lateral connectivity between both excitatory and inhibitory neurons, and additionally in Layer 2/3 we also model long range excitatory connections onto other excitatory and inhibitory neurons [18, 20, 125] (see Fig 1A and 1B). Layer 4 excitatory neurons send narrow projections to Layer 2/3 neurons (see Fig 1E). In addition to the detailed description of the model provided in this section, we have also included an overview of the model's specification with tables following the template proposed by Nordlie *et al.* [126] (S1 Table).

**4.1.1 Neuron model.** All neurons are modeled as single-compartment exponential integrate-and-fire neurons (ExpIF; Eq 1). The time course of the membrane potential $V(t)$ of all modeled neurons is governed by:

$$\tau_{\mathrm{m}}\frac{\mathrm{d}V}{\mathrm{d}t} = -(V - E_{\mathrm{L}}) + \Delta_T\exp\left(\frac{V - V_T}{\Delta_T}\right) + R_{\mathrm{m}}g_{\mathrm{exc}}(E_{\mathrm{exc}} - V) + R_{\mathrm{m}}g_{\mathrm{inh}}(E_{\mathrm{inh}} - V) \quad (1)$$

where $g_{\mathrm{exc}}$ and $g_{\mathrm{inh}}$ are the incoming excitatory and inhibitory synaptic conductances (Section 4.1.6), and $V_T$ is the threshold parameter of the exponential action potential generation mechanism. Spikes are registered when the membrane potential crosses -40 mV, at which time the membrane potential is set to the reset value $V_{\mathrm{r}}$ of -60 mV. Each spike is followed by a refractory period during which the membrane potential is held at $V_{\mathrm{r}}$. For excitatory neurons, $E_{\mathrm{L}}$ and $V_T$ were respectively set to -80 mV and -57 mV; for inhibitory neurons, they were set to -78 mV and -58 mV. The membrane resistance in cat V1 in the absence of synaptic activity has been estimated to be on average $\sim 250$ M$\Omega$ for excitatory neurons, and that of inhibitory neurons has been estimated to be higher [44]. To reflect these findings we set the membrane resistance of all cortical excitatory and inhibitory neurons $R_{\mathrm{m}}$ to 250 M$\Omega$ and 300 M$\Omega$ respectively. We set the membrane time constant $\tau_{\mathrm{m}}$ of excitatory neurons to 8 ms, and of inhibitory neurons to 9 ms. The refractory period is 2 ms and 0.5 ms for excitatory and inhibitory neurons respectively. Overall these neural parameter differences between excitatory and inhibitory neurons reflect the experimentally observed greater excitability and higher maximum sustained firing rates of inhibitory neurons [127]. The excitatory and inhibitory reversal potentials $E_{\mathrm{exc}}$ and $E_{\mathrm{inh}}$ are set to 0 mV and -80 mV respectively in accordance with values observed experimentally in cat V1 [44]. The threshold slope factor $\Delta_T$ was set to 0.8 mV for all neurons based on fits to pyramidal neuron data [128].

**4.1.2 Thalamo-cortical model pathway.** All neurons in the model Layer 4 receive connections from the model LGN (Section 4.2). For each neuron, the spatial pattern of thalamo-cortical connectivity is determined by the sum of a Gabor distribution (Eq 2), inducing the elementary RF properties in Layer 4 neurons [5, see Fig 1E and 1F], and a Gaussian distribution.

$$\begin{aligned} g(x, y, \lambda, \theta, \psi, \sigma, \gamma) &= \exp\left(\frac{x'^2 + y'^2\gamma^2}{2\sigma^2}\right)(G + \cos(2\pi x'\lambda + \psi)) \\ x' &= x\cos\theta + y\sin\theta \\ y' &= -x\sin\theta + y\cos\theta \end{aligned} \quad (2)$$

where $G$ is set at 0.085 and defines the relative weight of the Gaussian distribution with respect

to the Gabor distribution. For individual neurons the orientation $\theta$, phase $\psi$, size $\sigma$, frequency $\lambda$ and aspect ratio $\gamma$ of the distributions are selected as follows. To induce functional organization in the model, a pre-computed orientation map [129] corresponding to the $5.0 \times 5.0$ mm of simulated cortical area is overlaid onto the modeled cortical surface (see Fig 2A), thereby assigning each neuron an orientation preference $\theta$ (see Fig 2B). For the sake of simplicity, the phase $\psi$ of the Gabor distribution is assigned randomly (but see Section 3.6). The remaining parameters are set to constant values, matching the average of measurements in cat V1 RFs located in the para-foveal area. Taking into consideration the estimated mean RF diameter at para-foveal eccentricities [61], and assuming that the measured RF perimeter corresponds to the $3\sigma$ envelope of the Gabor function, we set the size $\sigma$ parameter to 0.17 degrees of visual field. The spatial frequency $\lambda$ was set to 0.8 cycles per degree based on experimental measurements [130]. Finally, studies have produced wide range of estimates for the mean of the RF aspect ratio, ranging from 1.7 to 5 [131]. We have chosen to set the value of the $\gamma$ parameter to 2.5, which is at the lower end of this range, as studies restricted to only layer 4 cells produced such lower estimates.

**4.1.3 Cortico-cortical connectivity.** The number of synaptic inputs per neuron in cat V1 has been estimated to be approximately 5800 [132]. While a substantial portion of these synapses come from outside V1, the exact number has not yet been established. A recent investigation by [96] found that even for a cortical section 800μm in diameter 76% of synapses come from outside of this region, while a rapid falloff of bouton density with radial distance from the soma has been demonstrated in cat V1 [18]. Furthermore, V1 receives direct input from a number of higher cortical areas and sub-cortical structures. Feedback from V2 alone accounts for as many as 6% of synapses in supra-granular layers of macaque [104]. Altogether it is reasonable to extrapolate from these numbers that as many as 50% of the synapses in layers 4 and 2/3 originate outside of V1. Furthermore, cortical synapses have been shown to frequently fail to transmit arriving action potentials, with typically every other pre-synaptic action potential failing to evoke a post-synaptic potential [133, 134]. Because it would be computationally expensive to explicitly model the synaptic failures, to account for the resulting loss of synaptic drive, we have factored it into the number of simulated synapses per neuron. Thus, the average number of synaptic inputs (5800), the estimate of the proportion of extra-areal input (50%) and the failure rates of synaptic transmission (50%) would yield approximately 1450 synapses per layer 4 neuron. To keep the computational complexity of the simulations manageable, while keeping these estimates in mind we decided to model 1000 reliable synaptic inputs per modeled excitatory cell in Layer 4. In line with the observations from Binzegger *et al.* [19] (reported by Izhikevich *et al.* [135]) that Layer 2/3 has more numerous recurrent connections than Layer 4, each modeled excitatory cell in Layer 2/3 receives 2300 synaptic inputs. Inhibitory neurons receive 40% fewer synapses than their excitatory counterparts, to account for their smaller size, but otherwise synapses are formed proportionally to the two cell type densities. 22% of the Layer 2/3 neurons synapses originate from Layer 4 cells [135]. In addition Layer 4 cells from each excitatory and inhibitory population receive additional thalamo-cortical synapses from a uniform distribution with respective boundaries of [90, 190], and [112, 168], which lies within the range of experimental observations [29]. The synapses are drawn probabilistically with replacement (with functional and geometrical biases described below), with self-connections being allowed. However, because we allow formation of multiple synapses between neurons, the exact number of connected neurons and the effective unitary strength of these connections is variable.

The geometry of the cortico-cortical connectivity is determined based on two main principles: (i) the connection probability falls off with increasing cortical distance between neurons [18, 20, 136] (see Fig 1A and 1B), and (ii) connections have a functionally specific bias [18, 90]. The two principles are each expressed as a connection-probability density function, then

multiplied and re-normalized to obtain the final connection probability profiles, from which the actual cortico-cortical synapses are drawn. The following two sections describe how the two probability density profiles of connectivity are obtained.

Finally, apart from the connectivity directly derived from experimental data, we also modeled a direct feedback pathway from Layer 2/3 to Layer 4. Such direct connections from Layer 2/3 to Layer 4 are rare [19], however a strong feedback from Layer 2/3 reaching Layer 4 via layers 5 and 6 exists [19]. Since we found that closing of this strong cortico-cortical loop is important for correct expression of functional properties across the investigated layers, and because we do not explicitly model the sub-granular layers (Section 4.1), we decided to instead model a direct Layer 2/3 to Layer 4 pathway. For the sake of simplicity, we have assumed it has the same geometry as the feed-forward Layer 4 to Layer 2/3 pathway (see following two sections), and considered it represents 20% of the synapses formed in Layer 4.

**4.1.4 Spatial extent of local intra-cortical connectivity.** The exact parameters of the spatial extent of the model local connectivity, with the exception of excitatory lateral connections in Layer 2/3, were established based on a re-analysis of data from cat [20]. Let $M_{xyz}^{ij}$ be the probability of potential connectivity (Fig 7 in [20]), of a pre-synaptic neuron of type $i \in \{\text{exc, inh}\}$ at cortical depth $x$ to other post-synaptic neurons of type $j$ at cortical depth $y$ and lateral (radial) displacement $z$. We reduced the 3 dimensional matrix $M_{xyz}^{ij}$ to a single spatial profile for each pair of layers and neural types (excitatory and inhibitory). This was done as follows. For each possible projection $L_{\text{pre}}T_{\text{pre}} \rightarrow L_{\text{post}}T_{\text{post}}$ where $L$ corresponds to layers $L \in \{4, 2/3\}$ and $T$ corresponds to neuron type $T \in \{\text{exc, inh}\}$ we did the following:

1. Select section of $M_{xyz}^{T_{\text{pre}}T_{\text{post}}}$ along the depth dimension corresponding to the pre-synaptic layer $L_{\text{pre}}$ and post-synaptic layer $L_{\text{post}}$.

2. Average the selected section of $M$ along the the dimensions corresponding to pre- and post-synaptic layer depth, resulting in a vector representing the average lateral profile of potential connectivity of a neuron residing in layer $L_{\text{pre}}$ to neurons residing in layer $L_{\text{post}}$.

3. Normalize the resulting distance profile to obtain the probability density function.

To obtain a parametric representation of the distance connectivity profiles we fitted them with several probability distributions, including Gaussian, exponential and hyperbolic. We found that the best fits were obtained using the zero mean hyperbolic distribution:

$$pdf(x) = \exp(-\alpha\sqrt{\theta^2 + x^2}) \tag{3}$$

which was thus chosen as the parametric representation of the connectivity profiles. The resulting values of the parameters of the fitted hyperbolic distributions, for all combinations of pre- and post- synaptic layers and neuron types, which were used to generate the local connectivity distance dependent profiles in the model, can be found in Table 2.

This data [20] only reflects the local connectivity, due to its dependence on neural reconstruction in slices which cut off distal dendrites and axons further than $500\mu m$ from the cell body. In cat Layer 2/3 excitatory neurons send long-range axons up to several millimeters away [18, 125]. Following Buzás *et al.* [18] we model the lateral distribution of the Layer 2/3 excitatory connections as $G(\sigma_s) + \alpha G(\sigma_l)$, where $G$ is a zero mean normal distribution, $\sigma_s = 270$ $\mu$m and $\sigma_l = 1000$ $\mu$m are the short and long-range space constants chosen in-line with Buzás *et al.* [18], and $\alpha = 4$ is the ratio between the short-range and long-range components. Furthermore, the long-range connections in Layer 2/3 have like-to-like bias [18]. How we incorporate this functional bias is described in the following section.

**Table 2. The parameters of hyperbolic profiles (Eq 3) of potential connectivity derived from [20] used to generate the distance dependent profile of local connectivity in the model.** The parameters for pathways not present in the model are not included in the table. Rows indicate pre-synaptic populations, whereas columns indicate post-synaptic populations.

| | $\alpha$ | | | | $\theta$ | | | |
|---|---|---|---|---|---|---|---|---|
| | L4 Exc | L4 Inh | L23 Exc | L23 Inh | L4 Exc | L4 Inh | L23 Exc | L23 Inh |
| L4 Exc | 0.0139 | 0.0148 | 0.0174 | 0.0197 | 207.7 | 191.8 | 154.4 | 131.5 |
| L4 Inh | 0.0126 | 0.0119 | | | 237.5 | 256.4 | | |
| L23 Inh | | | 0.0149 | 0.0150 | | | 189.5 | 188.61 |

**4.1.5 Functionally specific connectivity.** The functionally specific connectivity in V1 is poorly understood. Experimental studies have shown that local connections in cat are diffuse, without a strong orientation bias [18], while long-range connections have a moderate bias towards iso-orientation configurations [18, 95]. Recent more detailed studies of local connectivity in mice have shown that local connections also have a weak bias towards connecting neurons with similar receptive field properties [90, 137]. Furthermore, the anti-phase relationship between excitatory and inhibitory synaptic conductances found in a majority of cat V1 Simple cells [21, 97] can be interpreted as indirect evidence for inhibitory inputs to be biased towards neurons with anti-correlated RFs (i.e. co-tuned but anti-phase; although inhibitory conductance shows greater diversity of orientation tuning relative to spike-based preference [39, 44, 93]). Overall, these anatomical and functional findings point to a weak tendency of excitatory neurons towards connecting nearby neurons of similar RF properties while this bias increases somewhat for more distant post-synaptic neurons, and within Layer 4 Simple cells for inhibitory neurons to exhibit weak bias towards connecting to neurons with anti-correlated RFs.

Within Layer 4, the above findings point to the push-pull scheme of connectivity that has been demonstrated to explain a range of phenomena related to orientation tuning [5] (see Fig 1E). To implement this connectivity in our model we performed the following process. For each pair of Layer 4 neurons the correlation $c$ between their afferent RFs (i.e. RF defined only by the afferent inputs from thalamus) is calculated. The connectivity likelihood for a given pair of neurons is then given by $\frac{1}{\sigma\sqrt{2\pi}}e^{-(c-\mu)^2/2\sigma^2}$ where $\sigma = 1.3$, and $\mu = 1$ if the pre-synaptic neuron is excitatory or $\mu = -1$ if it is inhibitory.

In Layer 2/3 excitatory neurons send long-range connections spanning up to 6 mm to both excitatory and inhibitory neurons, preferentially targeting those with similar orientation preference [18]. To reflect this connectivity in the model we define the functional connectivity likelihood between pairs of neurons in Layer 2/3 as $\frac{1}{\sigma\sqrt{2\pi}}e^{-(\Delta o)^2/2\sigma^2}$ where the $\Delta o$ is the difference between the orientation preference of the two neurons, and $\sigma$ is set to 1.3 rad. The connectivity likelihoods described above are renormalized for each neuron to obtain probability density functions. Note that only the long-range component of the Layer 2/3 model is multiplied by this functionally specific bias, while the short-range component remains non-specific (Section 4.1.4). Overall this parameterization leads to only weak bias towards co-tuned connections in both simulated cortical layers.

**4.1.6 Synapses.** The relatively few studies that have examined in detail the strength of cortical synapses have found that unitary synaptic strengths are generally weak ($<2$ nS), but broad ranges have been reported [138, 139]. While the overall specificity of the synaptic strength between pairs of neural types or layers remains unclear, a recent study in cat V1 has shown that synapses from excitatory onto inhibitory cells are stronger than those targeting excitatory neurons [138, 139]. Reflecting these insufficient experimental constraints, the synaptic weights were selected to achieve an overall balance between excitation and inhibition

that supports reasonable levels of both spontaneous and evoked activity, while being compatible with the limited physiological findings. Specifically, we set the Layer 4 intra-laminar excitatory-to-excitatory synapses to 0.18 nS and excitatory-to-inhibitory synapses to 0.22 nS, while inhibitory synapses are set to 1 nS. In Layer 2/3, the synaptic weights were set to the same value as Layer 4 with the exception of the excitatory-to-inhibitory synaptic weights which were set to 0.35 nS. The thalamo-cortical synapses were set to 1.2 nS, reflecting the findings that these synapses tend to be larger [29] and more reliable [133] than their intra-laminar counterparts. The weights of synapses formed by connections from Layer 4 to Layer 2/3 were set to 1 nS. The data concerning the strength of feedforward inter-laminar synapses relative to intra-laminar synapses is lacking in V1, and we therefore acknowledge that this value might be too high. Nevertheless, such strong feedforward synapses to Layer 2/3 were necessary to drive these neurons sufficiently to compensate for the absence of feedback connections from higher visual area. The weights of synapses formed by the feedback connection from Layer 2/3 to Layer 4 were set to the same value as for the excitatory intra-laminar Layer 4 synapses. Unitary synaptic inputs are modeled as transient conductance changes, with exponential decay with time-constant $\tau_e$ = 1.5 ms for excitatory synapses and $\tau_i$ = 4.2 ms for inhibitory synapses.

We have also modeled synaptic depression for thalamo-cortical, and excitatory cortico-cortical synapses [140] using the model of [141]. For the thalamo-cortical synapses we assume parameters corresponding to strong depression similar to [142] and [65] (U = 0.75, $\tau_{rec}$ = 125 ms, $\tau_{psc}$ = 1.5 ms and $\tau_{fac}$ = 0 ms). For the cortico-cortical excitatory synapses we assumed moderate depression (U = 0.75, $\tau_{rec}$ = 30 ms, $\tau_{psc}$ = 1.5 ms and $\tau_{fac}$ = 0 ms), in line with [141], except for the feeedback synapses for which we assumed weaker depression (U = 0.75, $\tau_{rec}$ = 20 ms, $\tau_{psc}$ = 1.5 ms and $\tau_{fac}$ = 0 ms). The inhibitory neurons in our model correspond mostly to fast-spiking basket cells, as they are the main source of inhibition in V1. Basket cells in Layer 2/3 form depressing synapses with both excitatory [143] and inhibitory [144] neurons in the neocortex of rodents, although this has not been yet quantified in cats to the best of our knowledge. For Layer 4 inhibitory synapses, we decided to use parameters corresponding to a stronger depression than for the cortico-cortical excitatory synapses (U = 0.75, $\tau_{rec}$ = 70 ms, $\tau_{psc}$ = 4.2 ms and $\tau_{fac}$ = 0 ms), while for the inhibitory synapses of Layer 2/3 we used the same recovery time constant as for the excitatory synapses (U = 0.75, $\tau_{rec}$ = 30 ms, $\tau_{psc}$ = 4.2 ms and $\tau_{fac}$ = 0 ms).

**4.1.7 Delays.**   We model two types of delays in the model circuitry. First, for all intra-cortical connectivity a distance-dependent delay with propagation constant of 0.3 ms$^{-1}$ [145–147] was used, which corresponds to the slow propagation of action potentials along the intra-V1 (lateral) un-myelinated axons. The delays in the feed-forward thalamo-cortical pathway are drawn from a uniform distribution within the range (1.4, 2.4) ms. Second, [148] have recently shown that delays of synaptic transmission in cat V1 are dependent on both pre- and post-synaptic neural type, with the notable feature of slow excitatory to excitatory and fast excitatory to inhibitory transmission. Distance-dependent axonal propagation delay is unlikely to explain these results as these experiments were performed in nearby neurons [148]. Thus, as suggested by [148], we have included a constant additive factor in all synaptic delays, specifically 1.4 ms for excitatory to excitatory synapses, 0.5 ms for excitatory to inhibitory synapses, 1.0 ms for inhibitory to excitatory synapses and 1.4 ms for inhibitory to inhibitory synapses, in line with the quantitative observations by [148]. We observed that the addition of this neuron-type-dependent delay factor improved the stability of the modeled cortical neural networks, reducing synchronous events during spontaneous activity. We hypothesized that this is due to the ability of inhibition to respond faster to any transient increase in activity in the network due to the shorter excitatory to inhibitory delay.

## 4.2 Input model

The input model described in this section corresponds to the whole retino-thalamic pathway. The cortical model corresponds to roughly $5.0 \times 5.0°$ of visual field (Fig 1C and 1F). To accommodate the full extents of the RFs of neurons at the edges of the model, the LGN model corresponds to $6 \times 6°$ of visual field. In the same manner, to accommodate the full extent of RFs of thalamic neurons the overall visual field from which the thalamic model receives input corresponds to $11 \times 11°$.

We simulate the responses of the LGN neurons using the canonical center-surround model of RF (Fig 1C). The centers of both ON and OFF LGN neurons RFs are uniformly randomly distributed in the visual space, with density 100 neurons per square degree. Each LGN neuron has a spatiotemporal RF, with a difference-of-Gaussians spatial profile and a bi-phasic temporal profile defined by a difference of gamma functions [149]. Due to the relatively small region of visual space our model covers, we do not model the systematic changes in RF parameters with foveal eccentricity and thus assume that all ON and OFF LGN neurons have identical parameters. The exact spatial and temporal parameters have been fitted to match the observations in [149].

To obtain the spiking output of a given LGN neuron, the visual stimulus, sampled into 7 ms frames, is convolved with its spatiotemporal RF. In addition, saturation of the LGN responses with respect to local contrast and luminance is modeled [3, 22]. For simplicity, the local luminance $ll$ is calculated as the mean of luminance values and local contrast $lc$ as the standard deviation of the luminances within the RF of a given neuron. The response of the linear RF is separated into a DC (luminance) component $r_l$ and a contrast component $r_c$. The saturation of the two components is modeled with two Naka-Rushton functions $\alpha_l \frac{r_l}{\beta_l + |r_l|}$ and $\alpha_c \frac{r_c}{\beta_c + |r_c|}$ respectively, where $\alpha$ is the gain and $\beta$ is the saturation parameter of the corresponding component. The parameters $\alpha$ and $\beta$ were empirically adjusted to obtain luminance and contrast response curves whose saturation point and level are within the ranges observed experimentally [3, 22] The values of each parameter of the input model can be found in S4 Table.

The resulting luminance and contrast temporal traces are then summed and injected into integrate-and-fire neurons as a current, inducing stimulus dependent spiking responses. In addition to the stimulus-dependent drive, each LGN neuron is also injected with an independent white noise current. The magnitude and variance of this noise is such that ON and OFF LGN neurons fire at $\sim 17$ and $\sim 8$ spikes/s respectively in the spontaneous condition [22].

## 4.3 Stimulation protocols

Each stimulation protocol consists of a series of visual stimuli which are interleaved with 150 ms of 50 cd/m$^2$ gray blank stimuli. The population of recorded neurons, with the exception of the size tuning protocol (Section 4.3.2), was always restricted to the central circular patch (radius of 1 mm) of the model to avoid contamination by potential edge-effects due to the finite simulated cortical area. Because the recording of intracellular signals at high temporal resolution for such a long stimulus set as presented in this work is extremely memory consuming, we recorded the membrane potential and excitatory and inhibitory conductances only from a subset of 190 neurons confined to the central $0.2 \times 0.2$ mm patch of the simulated cortical space, and whose orientation preference was within 0.25 radians of horizontal based on their position on the pre-computed orientation map. The durations of all visual stimuli are aligned with the 7 ms frame duration at which the retino-thalamic model operates (Section 4.2). Within each experimental protocol, stimuli were presented in a random order.

**4.3.1 Drifting sinusoidal grating protocol.** The spatial and temporal frequency of the RFs of the modeled LGN neurons (Section 4.1.2), and of the Gabor distribution template from which thalamo-cortical synapses were sampled, were identical for all simulated LGN and cortical neurons respectively. An important consequence of this simplification is that it allowed us to efficiently execute protocols requiring DGs. By employing a full-field stimulus with spatial frequency matching the Gabor template (0.8 Hz) and drifting at 2 Hz, we were in parallel stimulating all cortical neurons with a stimulus with optimal spatial frequency. We varied the orientation of the gratings between 0 and 180 degrees, in 8 equal steps. We used 3 values of contrast: 10%, 30% and 100%. Each grating was shown 10 times for 2002 ms. We recorded spikes from 2025 randomly selected neurons for this experiment.

**4.3.2 Size tuning protocol.** Size tuning was measured using DGs of optimal spatial and temporal frequency (see orientation tuning protocol for details), confined to an aperture of variable diameter. The orientation of the gratings was horizontal (0˚) and the position was in the center of the simulated visual field. The diameter of the aperture was varying between 0˚ and 10˚ in 12 steps, with contrast of either 10% or 100%, each presentation lasting 2002 ms and being repeated 10 times.

The center of the modeled cortical area was occupied by a horizontal orientation domain (see Fig 2). During the size tuning protocol we recorded from neurons in this central (horizontal) orientation domain and selected neurons whose RFs centers were within 0.4˚ from the center of the simulated visual field and whose orientation preference was within 0.25 radians from horizontal based on their positions on the pre-generated orientation map. This setup allowed us to simultaneously determine the size tuning of the population of neurons reported here, with precision comparable to experimental studies, while greatly reducing the computational resources required for the simulation.

**4.3.3 Natural images with simulated eye-movement protocol.** We have replicated the natural image with simulated eye-movement (NI) protocol introduced by Baudot *et al.* [21]. This protocol emulates the global retinal flow experienced during the exploration of the natural environment by simulating the retinal impact of visuomotor interaction. This is done by imposing shifts and drifts of a static natural scene that reproduce the kinematics of a realistic ocular scanpath [21] recorded in the intact behaving animal. We used the same image as in Baudot *et al.* [21], scaled to match the size of the simulated visual space, and the same path. Presentation of the resulting movie lasting 2002 ms was repeated 10 times. We recorded spikes from 2025 randomly selected neurons for this protocol.

## 4.4 Data analysis

Unless specified otherwise, all tuning curves were calculated as the trial averaged mean firing rate during stimulation with the given stimulus and parameter value (e.g. orientation of sinusoidal grating). Spontaneous activity level was not subtracted. All analog signals were recorded at 1 ms resolution.

We calculated the *irregularity* of an individual neuron's spike train as the coefficient of variation (CV) of its inter-spike-intervals (ISI):

$$CV_{ISI}^2 = \frac{\mathrm{Var}[ISI]}{\langle ISI \rangle^2} \qquad (4)$$

where Var corresponds to variance, and $\langle \rangle$ is the mean. To ensure accurate estimation of the irregularity we exclude neurons that fired fewer than 10 spikes during the recording. We consider the threshold for irregular firing to be $CV_{ISI} > 0.9$ [45]. We assess synchrony in our network by calculating the average cross-correlation (CC) of spike histograms $SC_i$ $SC_j$ between

disjoint pairs of all recorded neurons $i$, $j$.

$$CC[SC_i, SC_j] = \frac{\text{Cov}(SC_i, SC_j)}{\sqrt{\text{Var}(SC_i)\text{Var}(SC_j)}} \tag{5}$$

where Var is again variance and Cov is covariance of the spike count. The spike counts were calculated by counting the number of spikes within 10 ms bins.

In order to assess orientation tuning, we calculated two complementary measures: the half width at half height (HWHH) and relative unselective response amplitude (RURA) [48]. To calculate HWHH we fitted the orientation tuning curves with a Gaussian function [150]:

$$R(\phi) = \beta + \alpha \exp\left(\frac{(\phi - \phi_{\text{pref}})^2}{2\sigma^2}\right) \tag{6}$$

where $R$ is the spiking response of the given neuron to a sinusoidal grating with orientation $\phi$, $\phi_{\text{pref}}$ is the preferred orientation of the given neuron, $\sigma$ is the width of the tuning, $\beta$ is the baseline activity and $\alpha$ a scale factor. Low responding neurons (less then 1 spike/s at optimal orientation) were excluded from the analysis, as reliable curve fitting was not possible with the amount of recorded data. Furthermore, neurons for which a reliable fit of the Gaussian curve was not possible ($MSE > 30\%$ of the tuning curve variance) were also excluded from this analysis. In total, a minority consisting of 7% of the neurons was excluded. HWHH was then calculated as $\sqrt{2\ln 2}\sigma$. RURA was calculated from the fitted Gaussian parameters according to the following equation [150]:

$$RURA = \frac{\beta}{\beta + \alpha} \tag{7}$$

The modulation ratio (MR) of spike responses was computed as the ratio of the fundamental Fourier component F1 and the DC component F0 of the neuron's peri-stimulus time histogram (PSTH) in response to a DG [151]. The F1 component was defined as the Fourier component corresponding to the temporal frequency of the DG stimulus. The PSTH was formed with ms bins, and spontaneous activity was subtracted prior to MR calculation. Cells with MR greater than one are classified as Simple cells, cells with MR lower than one are classified as Complex cells. The MR of the membrane potential was calculated analogously as the F1/F0 component ratio of the membrane potential after subtraction of its resting level.

To quantify the parameters of size tuning curves, we first fitted the data with the Supplementary Difference of Gaussians (3G) model [55] defined as:

$$f(r) = K_c G_c(r) - K_s G_s(r) + K_{cs} G_{cs}(r) \tag{8}$$

where

$$G_x(d) = \left(\frac{2}{\sqrt{\pi}} \int_0^d e^{-(y/w_x)^2} dy\right)^2 \tag{9}$$

where $d$ is the stimulus diameter, $K_c$ is the strength of the center, $K_s$ is the strength of the suppressive surround, $K_{cs}$ is the strength of the counter-suppressive surround and $w_x$ are the space constants of the corresponding terms. We then performed the following analysis on the fitted curves. We excluded low responding neurons (response of less then 2 spikes/s at the optimum size) from the analysis as well as neurons for which a reliable fit was not possible ($MSE > 30\%$), leading to 31% of the neurons being discarded. We defined the summation

peak $\delta_{max}$ as the diameter where maximum response is achieved before suppression sets in (we also refer to this diameter as the maximum facilitation diameter (MFD)), suppression peak $\delta_{min}$ as the diameter where the minimum response is achieved across all aperture sizes greater than $\delta_{max}$, and the counter-suppression peak $\delta_{cs}$ as the diameter where the maximum response is achieved across all aperture sizes greater than $\delta_{min}$. This in turn allows us to define the peak summation response $R_{max}$, peak suppression response $R_{min}$, and peak counter-suppression response $R_{cs}$ as the neuron's responses at the corresponding aperture sizes. We can then define the suppression index (SI) analogously to experimental studies [55] as:

$$SI = \frac{R_{max} - R_{min}}{R_{max}} \tag{10}$$

and the counter-suppression index CSI as:

$$CSI = \frac{R_{cs} - R_{min}}{R_{max}} \tag{11}$$

The reliability and the precision of the responses were measured by fitting a Gaussian function to the mean cross-correlation across trials (Eq 5) of the spiking responses, and of the sub-threshold membrane potential responses [21]. Spikes were removed from the membrane potential prior to the analysis, by replacing the membrane potential within a 5 ms window centered on each spike with the signal interpolated linearly between the levels of membrane potential before and after the window. The reliability was then defined as the CC peak amplitude at time zero, and the temporal precision by the standard deviation of the Gaussian fit.

Quantitative values in this article are expressed as mean ± SEM.

## Supporting information

**S1 Fig. Decrease of synaptic input in Layer 4 in the absence of cortico-cortical connections.** (A) Total synaptic conductance for an example Layer 4 excitatory cell during spontaneous activity in our V1 model (orange) and in a version containing only the thalamo-cortical connections to Layer 4 (blue). (B) Same, for evoked activity in response to a high contrast drifting grating with an orientation similar to the preferred orientation of the cell, averaged over 10 trials. (C) Same as (A), but for an example inhibitory neurons of the Layer 4 of the model. (D) Same as (B), but for an example inhibitory neurons of the Layer 4 of the model. (E) Layer 4 average total synaptic conductance for 40 seconds of spontaneous activity, for both our V1 model (orange) and the version with only cortico-thalamic afferents (blue). (F) Same as (E), for a subpopulation of cells that have corresponding preferred orientations, and for evoked activity in response to a high contrast drifting grating with an orientation similar to the preferred orientation of the cells, averaged over 10 trials.
(PNG)

**S2 Fig. The inverse of the stimulus-locked time-averaged standard deviation of membrane potential (1/SD) across trials, averaged across cells, relative to the value for ongoing activity.** Results are averaged across all recorded excitatory neurons pooled across the two model layers. (A) For our V1 model with no external noise other brain areas. (B) For a modified version of our V1 model with 100 Hz of external noise input to every cortical neuron, with the weight of synapses formed by external connections to cortex set to 0.9 nS, and the weight of synapses formed by connections from Layer 4 to Layer 2/3 set to 0.9 nS. (C) Same as B with 1000 Hz of external noise, with the weight of synapses formed by external connections to cortex set to 0.65 nS, the weight of thalamo-cortical connections set to 0.675 nS and the weight of synapses formed by connections from Layer 4 to Layer 2/3 set to 0.65 nS. (D) Experimental

results in V1 of the anesthetized cat [21].
(PNG)

**S1 Table. Tabular description of the model, part 1.** The model is summarized in panel A and described in more details in panels B-G. See S2 Table for panel D, S3 Table for panel E, S4 Table for panel F, and S5 Table for panel G.
(PNG)

**S2 Table. Tabular description of the model, part 2.** See S1 Table for panels A-C, S3 Table for panel E, S4 Table for panel F, and S5 Table for panel G.
(PNG)

**S3 Table. Tabular description of the model, part 3.** See S1 Table for panels A-C, S2 Table for panel D, S4 Table for panel F, and S5 Table for panel G.
(PNG)

**S4 Table. Tabular description of the model, part 4.** See S1 Table for panels A-C, S2 Table for panel D, S3 Table for panel E and S5 Table for panel G.
(PNG)

**S5 Table. Tabular description of the model, part 5.** See S1 Table for panels A-C, S2 Table for panel D, S3 Table for panel E and S4 Table for panel F.
(PNG)

## Author Contributions

**Conceptualization:** Ján Antolík, Cyril Monier, Yves Frégnac, Andrew P. Davison.

**Formal analysis:** Ján Antolík, Rémy Cagnol.

**Funding acquisition:** Ján Antolík, Cyril Monier, Yves Frégnac, Andrew P. Davison.

**Investigation:** Ján Antolík, Rémy Cagnol, Tibor Rózsa.

**Methodology:** Ján Antolík, Rémy Cagnol.

**Software:** Ján Antolík, Rémy Cagnol, Tibor Rózsa, Andrew P. Davison.

**Supervision:** Ján Antolík, Cyril Monier, Yves Frégnac, Andrew P. Davison.

**Validation:** Ján Antolík, Rémy Cagnol.

**Visualization:** Ján Antolík, Rémy Cagnol.

**Writing – original draft:** Ján Antolík.

**Writing – review & editing:** Ján Antolík, Rémy Cagnol, Cyril Monier, Yves Frégnac, Andrew P. Davison.

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
