## [Decision Letter · Decision Letter 0]

15 Feb 2024

Dear Antolik,

Thank you very much for submitting your manuscript "A comprehensive data-driven model of cat primary visual cortex" for consideration at PLOS Computational Biology.

As with all papers reviewed by the journal, your manuscript was reviewed by members of the editorial board and by several independent reviewers. In light of the reviews (below this email), we would like to invite the resubmission of a significantly-revised version that takes into account the reviewers' comments.

The three reviewers commend the quality of the results presented in the manuscript. However, they raise important points that must be addressed, should you decide to submit a revised version. Among their comments, we agree that it is important for the impact of the article that the code of the model be available in a form that makes it easy to use by the community. This means both that the totality of the frameworks used should be available and that the documentation of the code should be substantial. The reviewers also point important aspects of the modelling itself, for instance regarding the strength of the internal drive and the absence of stochastic background input, as well as the width of the orientation tuning curves compared to the experimental literature.   

We cannot make any decision about publication until we have seen the revised manuscript and your response to the reviewers' comments. Your revised manuscript is also likely to be sent to reviewers for further evaluation.

Sincerely,

Hugues Berry

Academic Editor

PLOS Computational Biology

Thomas Serre

Section Editor

PLOS Computational Biology

Reviewer's Responses to Questions

**Comments to the Authors:**

Reviewer #1: In the study "A comprehensive data-driven model of cat primary visual cortex", Antolik et al. present a spiking neural network models of layers 2/3 and 4 of the central part of cat primary visual cortex. The model neurons are connected in a biologically plausible way based on anatomical connectivity studies. This includes unspecific spatially dependent connectivity, patchy connectivity based on an orientation tuning map, and push-pull connectivity. Additionally, an input model is devised to provide the modeled tissue with visual input via the LGN. The authors investigate the dynamical behavior of their model under a variety of stimulation protocols, including drifting gratings and natural images. They find a that the model can, among other things, qualitatively or quantitatively reproduce spontaneous activity with plausible firing rates for excitatory and inhibitory neurons, contrast-invariant orientation tuning curves for single neurons, emergence of simple and complex cells separated into the two layers, surround suppression, and differences in trial-to-trial variability for drifting gratings or natural images. This is embedded into a larger suite of validation tests that constrain the model and show its ability to reproduce physiologically plausible behavior in a variety of scenarios. Integrative modeling like that presented here is an important endeavor towards combining the available knowledge on structure and function in a coherent framework. The model accounts for an impressive range of findings on the early visual system. However, there are also some unrealistic aspects that are glossed over or even presented as being realistic (details below). Most importantly, the issue of the missing external drive (presented as if it were a strength rather than a limitation) should be adequately addressed. Being open-source, the model can be built on by others, although the documentation should be improved for this purpose.

General

* It would aid the understanding of the model specification if the authors could provide tables in the style of Nordlie et al. 2009, especially for connectivity and single-neuron parameters. It should also be specified if self-connections are allowed.

* In several places, the PSTH is used for ongoing activity, seemingly not aligned to a stimulus. Perhaps this would be better just called 'binned spike trains', 'spike histogram', or similar.

* The authors should discuss limitations due to the downscaling of the numbers of neurons and synapses with respect to real brains. With the assumptions of 50% local connectivity (which is rather low) and 50% failure rate, one still arrives at 1450 synapses per neuron, of which only 1000 are modeled.

* The classification into simple and complex cells does not consider the main properties that distinguish these cells types: larger receptive fields of complex cells and a lack of clear on and off regions for complex cells.

* The README in the repository https://github.com/CSNG-MFF/mozaik-models/tree/main/LSV1M is rather minimal and should contain an overview of the files included in the repository and what each file is for. I was for instance not able to identify the precomputed orientation map that was used. There are two candidates with apparently different resolutions (or_map_new_16x16 and or_map_new_6x6) but the documentation is missing. A file whose name suggests it is a main file, run.py, appears to be about a different model, that of Vogels and Abbott. Furthermore, the required software/package versions should be mentioned.

* The figures have very low resolution; presumably this would be fixed if the paper is accepted.

Introduction

l. 43: "Thalamocortical connectivity in Layer 4 follows a push-pull organization [...] .": While the push-pull connectivity is mediated by thalamocortical projections, it is itself a characterization of the intra-cortical connectivity within L4 of V1, and not of the thalamocortical connectivity targeting Layer 4. The above sentences makes this sound otherwise.

Results

l. 75: "The only external source of variability in the model is a white noise current injection into lateral geniculate nuclesu (LGN) cells which induces their spontaneous firing rate at about 17 Hz for ON cells and 8 Hz for off cells [...]".

As a part of the cortex, the modeled patch would not only receive input from parts of V1 not modeled, but also from other cortical areas. Especially the foveal region in the area centralis is known to be strongly targeted by feedback projections, at least in macaque V1 (Wang et al. 2022, Retinotopic organization of feedback projections in primate early visual cortex). While modeling other parts of the primary visual cortex not represented in the given model (other layers for example, or areas establishing feedback projections to V1) is clearly beyond the scope of the presented work, they still could be provided as stochastic 'background input'. Why is this not done?

l. 80: "Due to the low proportion of synapses of thalamic origin in the model cortex, the variability of spontaneous and visually evoked cortical activity is largely driven by internal cortical dynamics."

This is asserted without evidence. Surely, since neurons in L2/3 do not receive external input from outside the model this is true trivially. But for neurons in L4 one can do the following back-of-the-envelope calculation:

In the model, L4 excitatory cells have 1000 recurrent and around 140 TC synapses. Let's assume that 50% of synapses are excitatory, 50% inhibitory (probably there are more excitatory).

Spontaneous firing rate for excitatory neurons in L4 is 1.4 Hz, in L2/3 2 Hz. The weight for L4E->L4E connections is 0.18 nS, for L2/3E->L4E 1 nS.

The latter connections are 20% of the synapses formed there, so 200 per cell.

This implies that the incoming recurrent input per excitatory neuron from all excitatory neurons is about (omitting units) 500 * 0.8 * 1.4 * 0.18 + 500 * 0.2 * 2 * 1 = 100.8 + 200 = 300.

The inhibitory firing rate is 7.6 spikes/s in L4, 4.7 spikes/s in L2/3. The inhibitory weights are set to 1 nS.

This results in incoming recurrent input per excitatory neuron from inhibitory neurons (again omitting units, ignoring different signs from excitatory and inhibitory neurons) 500 * 0.8 * 7.6 * 1 + 500 * 0.2 * 4.7 * 1 = 3040 + 470 = 3500.

There are 140 TC synapses per neuron firing at a rate of about 12 spikes/s with a weight of 1.2 nS leading to an input of about 140 * 12 * 1.2 = 2016.

The fraction of TC input to excitatory cells now is 2016 / (3500 + 300 + 2016) = 0.35.

This fraction gets higher if there are relatively more excitatory recurrent connections, so that it may actually exceed 0.5. Furthermore, synchronization plays a role in determining which connections are more influential. It is thus unclear if indeed "variability of spontaneous and visually evoked activity is largely driven by internal cortical dynamics."

Fig. 3B-E: The red traces are not sufficiently clearly visible. The same holds for some of the red traces in Fig. 5.

l. 85: In panels A-F of Figure 4, the authors always plot quantities for L2/3 neurons above the same data for L4 neurons. In panel G, however, neurons in L4 are above L2/3. This then is consistent with Figures 5, 6, 7, 8, but not with the raster plot. Please make this consistent to increase understandability.

l. 90-91: It would be better to compare with awake recordings.

l. 97: It is very selective to compare with a study that reports higher spontaneous firing rates in L2/3 compared to L4. Many studies show the opposite (e.g., Gur et al., Cerebral Cortex 2005 for macaque V1; de Kock and Sakmann PNAS 2009 for rat S1; see also Potjans & Diesmann 2014).

l. 142: Please add references on experimental studies showing steady depolarization in L2/3 neurons in response to optimally oriented gratings.

l. 150: It seems that this is the first mention of Figure 6. It starts with panel C, not panel A. This is a bit confusing. It would be better to refer to panel A first or change the order of panels within the figure.

Figure 6 (correctly) uses sp/s as the unit for the firing rate, whereas in other places, 'Hz' is used, which is actually reserved for processes with a fixed frequency.

Fig. 6: It is strange that the integrals of the histograms in corresponding lines of panels C and D of Fig. 6 do not amount to the same numbers of neurons. It seems these panels do not actually share the same y-axis.

Figures 6 and 7: It is highly unrealistic to have lower contrast leading to greater responses (see for instance Albrecht and Hamilton J Neurophysiol 1982; Reich et al. J Physiol 2000). I suspect the legends are reversed (as also suggested by the Fig. 7 caption, which says the blue lines are for high contrast and the black lines for low contrast). Also please clarify if the low contrast curves in these figures are for 10% or for 15% contrast, as conflicting information is given (also in Fig. 9). Finally, please mention in the Fig. 7 caption what is the meaning of the shaded areas, and make sure that the shading is consistently applied across all panels.

l. 190: "To assess to what extent this idealized scheme is true in this model, we have plotted the orientation tuning curves of the mean (DC) and first harmonic (F1) components of the sub-threshold signals across the different neural types and layers in Fig 7." Here "DC" is used as abbreviation, whereas the figure and caption use "F0". Please harmonize the notation here. Also, please clarify how F1 is computed.

l. 202: I'm not sure "explains" is the right word here.

l. 245-246: Please give units for the F1 components.

Figure 9: Please add units to horizontal axes at lowest plot in panel A, B, C.

Panel G: What is CRF? Classical receptive field? I think the abbreviation is not introduced.

l. 256: "While most excitatory neurons in both cortical layers show surround suppression [...]." This seems not to be quantified. In the caption of Figure 9 it says that traces of typical cells in L4 and L2/3 are shown. How many cells are "typical"?

l. 284: Instead of the central eccentricity, it makes more sense to compare with the average eccentricity of the model here (which will also be smaller than the ones in the experiments).

Figure 10: Figure 10B is referenced before 10A. It would be better to swap them. It would also be interesting to see the same plot for L2/3.

l. 327: "slightly higher": Here, 0.63 is called "slightly higher" than 0.441, whereas the previous sentence called 0.221 simply "higher" than 0.187. The former difference is greater in both absolute and relative terms; the wording should not try to mask this.

l. 344: "The lack of direct external noise injected into cortical model neurons was crucial for achieving the results presented in this section." The authors state that another model of V1 could not reproduce the stimulus-dependent variability effect at the level of the membrane potential, albeit being able to do so on the level of the spiking activity. It is argued that this is due to the lack of external input noise injected into the model neurons. It seems plausible, however, that neurons in V1 receive input from the surrounding cortex as well as other cortical areas that might - at least to some degree - 'look' like noise. What is the justification for not adding this, apart from thereby obtaining the seemingly correct result? If this only works in the absence of noise, something is probably wrong.

l. 336: The formulation calling the inverse of a standard deviation a "variance" is not ideal.

Discussion

l. 382: It is not at all a given that neuroscience should converge upon a single model. Even if one considers only a single individual of a single species, models at different levels of resolution and including different biological aspects will continue to be useful for different purposes.

l. 385: Similarly, it is not a given that all models should pass the same validation tests. First of all, experiments are not perfect and different experiments may even give contradictory results. Second, which validation tests a model should pass may depend on the purpose of the model.

Table 1: I am not sure if this can easily fit, but it would also be good to cross-link the validation tests the model passes with the respective figure/section where this is shown.

l. 536: "On the otger hand, [...]."

l. 582: What is meant by "inter-areal connectivity in V1"?

l. 631: "The model makes five main simplifications: [...]"

The subsampling of synapses is left out here.

Materials and Methods

l. 764: A spatial frequency should presumably be measured in inverse distance (or cycles per degree visual angle), not in Hz.

l. 778: "Altogether it is reasonable to extrapolate from these number that as many as 50% of the synapses in layers 4 and 2/3 originate outside of V1."

This seems implausible given that Markov et al. (2011) showed in a retrograde tracing study that after injection into macaque V1 around 80% of the source neurons in the hemisphere are inside that area. Even though neurons may establish different numbers of synapses, 50% from outside the area seems altogether too high. What is the exact basis of this assumption?

l. 787: "[...] rates of synaptic transmission (50%) [...]"

This seems to be taken from in vitro studies, but as the authors state themselves, the situation might be different in the presence of neuromodulators in vivo.

l. 857: "The functionally specific connectivity in V1 poorly understood."

This sentence lacks an "is".

l. 882: "In Layer 2/3 excitatory neurons send long-range [...]"

The functional connectivity likelihood between pairs of neurons in Layer 2/3 depends only on the difference in preferred orientation. Does this connectivity scheme have a connection probability that decreases with distance or is confined to a certain patch of cortical surface?

l. 885: The unit of sigma is missing.

l. 907: "The weights of synapses formed by connections from Layer 4 to Layer 2/3 were set to 1, while the weights of synapses formed by the feedback connection from Layer 4 to Layer 2/3 were set to the same value as for the excitatory intra-laminar synapses."

The sentence is broken: it refers to "Layer 4 to Layer 2/3" two times.

Also the synaptic weight lacks a unit.

Further, why is one type of intra-laminar connection so strong, 1 (nS, presumably), and the other so much weaker, ~ 0.2 nS?

l. 921: "Basket cells. 2'3 form [...]"

Broken notation.

l. 943: "We observed that the addition of this neuron-type-dependent delay factor improved the stability of the modeled cortical neural networks, reducing synchronous events during spontaneous activity."

This is not a new observation, see for example Brunel 2000.

l. 972: Please give the parameter values.

l. 1018: Was this the precomputed orientation preference or that measured in the model?

Eq. 6: It seems phi-phi_pref should be squared.

l. 1090: What were the specifics of the interpolation?

Reviewer #2: The paper reports on an impressive effort to combine many of the properties of the cat visual cortex, for which data is spread over several disparate sources, into a single, comprehensive model, and it argues convincingly for the importance of doing such detailed modeling work. The presentation of the model is well written, and readers will appreciate how clearly the model predictions are laid out, and how the experimental evidence, model assumptions and simplifications that went into constructing it are presented. I believe the paper will be valuable to anyone involved in simulating the visual cortex and for anyone interested in a mechanistic understanding of the various properties incorporated into the model. There is only one point in the paper that raised some concerns for me, detailed below.

The width of excitatory and inhibitory orientation tuning curves. In lines 154-155 and 413-414, the paper states that orientation tuning is only slightly broader for inhibitory neurons than it is for excitatory neurons. Focusing on L4 only, in the model the tuning HWHH is 21.1 vs 23.3 for excitatory and inhibitory neurons, so that inhibitory tuning curves are ~10% wider. But from what I could tell the sources indicate it should be much wider. The sources given were [8], [46], [47], and [65]. In source [47], Cardin et al 2007, they show in fig 4C and the corresponding text that in L4 the inhibitory tuning curves are ~5/3 times wider than excitatory ones on average. In sources [8] and [65], Finn et al 2007 and Nowak et al 2003, I could not find information specifically about the relative orientation tuning widths. In [46], Nowak et al 2007, they report in table 1 that inhibitory HWHH is ~4/3 times excitatory. That data is not layer specific, and they go on to speculate that the less selective inhibitory neurons they found came from L4, in which case the difference should be even larger in L4. If it is the case that I am reading these sources correctly, can you address the disparity between the model and data? Is it practical to tune the model parameters to match this data? If not, is it possible to identify why?

Very minor comments/suggestions below.

1. Small typos on lines 445, 536

2. Table 1. I believe readers will find table 1 and the corresponding text to be a useful resource for getting a birds-eye view of comparable efforts made to model the visual cortex and to see all of the properties integrated into the model of this paper. Since some of the models are based on primate V1, it may be useful to mark out in table 1 for which ones this is the case. The reason for this suggestion is that it is not clear to me that all of the properties listed in table 1 apply equally to cat and primate models. Additionally, should row 3, labeled "Log-normally distributed...", contain open red circles in the same places as row 5? Lastly, I found the label for row 6 slightly confusing because several of the models model spontaneous activity. If there is a more specific label to use in that row, it could improve the table.

Reviewer #3: The paper of Antolik et al. presents a large-scale spiking network model for cat primary visual cortex that goes beyond previous network studies in many ways and provides interesting results. The paper encapsulates extensive work, seems competently done, and provides a new important contribution to the field. I am therefore positive to publication of the paper in the journal.

My only major comment relates to how this model can be used by the community. As I understand it the results presented in the paper all came from essentially the same set of (numerous) chosen model parameters. For the model to become a useful resource for the community, it is essential that users can test the model for other parameter choices.

While it is stated that the model will be available via v1model.arkheia.org, at the moment there is not much information here (only the figures shown in the paper, as I understand it). I assume that this will be updated in the process of publishing?

Further, the model is to be run via Mozaik, a tool that some of the authors have developed, but how this is done is not clear.

Maybe the authors could add a section illustrating the typical use of the model (for example, running it for new model parameters) for new users? Are there plans to make the model available via EBRAINS (where some of the authors are involved)?

One thing missing (as far as I can see) is the description of (i) on what computers the results were run and (ii) how long time it took to run the model on these computers.

Minor things:

- The recent paper by Billeh et al, Neuron (2020) on a comprehensive model for mouse visual cortex should be cited and the connection to the present work maybe discussed.

- In Table 1, specific references should be listed when referring to Wielard series and the other series.

- On line 536: otger -> other

- On line 857: "is" is missing

- On line 921: I guess "2´3" is a typo?

Must add info on

**Have the authors made all data and (if applicable) computational code underlying the findings in their manuscript fully available?**

Reviewer #1: Yes

Reviewer #2: Yes

Reviewer #3: **No: **The info on v1model.arkheia.org must be expanded

PLOS authors have the option to publish the peer review history of their article (what does this mean?). If published, this will include your full peer review and any attached files.

Reviewer #1: No

Reviewer #2: No

Reviewer #3: No
---

## [Decision Letter · Decision Letter 1]

28 Jun 2024

Dear Antolik,

Thank you very much for submitting your manuscript "A comprehensive data-driven model of cat primary visual cortex" for consideration at PLOS Computational Biology. As with all papers reviewed by the journal, your manuscript was reviewed by members of the editorial board and by several independent reviewers. The reviewers appreciated the attention to an important topic. Based on the reviews, we are likely to accept this manuscript for publication, providing that you modify the manuscript according to the review recommendations.

Reviewer#1 still has a handful of remarks that look easy to address: clarification of the caption for Table 2 and Table S1, lack of labels on the y-axis of Fig S2 or of an explanation regarding why the influence of the other visual areas is not accounted for, be it with a simple modelling choice like a stochastic input. Please do account for these remarks in your revised version.

Sincerely,

Hugues Berry

Section Editor

PLOS Computational Biology

Thomas Serre

Section Editor

PLOS Computational Biology

Reviewer's Responses to Questions

**Comments to the Authors:**

Reviewer #1: I am impressed by how much the manuscript has improved and now consider it basically fit for publication. In the following, I only respond to the main points (which have been satisfactorily addressed) and provide two new comments.

“It would aid the understanding of the model specification if the authors could provide tables in

the style of Nordlie et al. 2009, especially for connectivity and single-neuron parameters. It

should also be specified if self-connections are allowed.

We have added these tables ( Table S1-5) in the supplementary material. We have also added

the following sentence at lines 787-790: “In addition to the detailed description of the model

provided in this section, we have also included an overview of the model’s specification with

tables following the template proposed by Nordlie et al. (Nordlie et al. 2009) (Table S1).”

Our revised version now specifies that self-connections are allowed.”

This is a great improvement that makes it easier to get a comprehensive overview of the model.

########################################################################

“l. 75: "The only external source of variability in the model is a white noise current injection into

lateral geniculate nuclesu (LGN) cells which induces their spontaneous firing rate at about 17

Hz for ON cells and 8 Hz for off cells [...]".

As a part of the cortex, the modeled patch would not only receive input from parts of V1 not

modeled, but also from other cortical areas. Especially the foveal region in the area centralis is

known to be strongly targeted by feedback projections, at least in macaque V1 (Wang et al.

2022, Retinotopic organization of feedback projections in primate early visual cortex). While modeling other parts of the primary visual cortex not represented in the given model (other

layers for example, or areas establishing feedback projections to V1) is clearly beyond the

scope of the presented work, they still could be provided as stochastic 'background input'. Why

is this not done?

We have added the following two paragraphs into the discussion that address the reviewers

question:

“The absence of a significant portion of the incoming connections in our model (primarily due to

omission of extra-V1 innervation) raises several issues. To properly model all the connections

coming from extra-V1 areas would require us to model all those source areas, which is outside

of the scope of this study. This however raises the question of how to compensate for the

absence of the drive from the missing connections. A common solution is to model an extra

independent Poisson spiking input into each cortical neuron (Schmidt et al. 2018), that

compensates for the missing drive. This is quite reasonable when the rate of such

compensatory input can be estimated as approximately Poissonian (e.g. spontaneous

condition). However, for visually evoked conditions, where the magnitude of the external drive

might depend on the origin of its source, the ongoing visual stimulus and likely even the specific

function of the receiving V1 neuron, constructing a reasonable approximation of an extra-areal

drive is extremely difficult. Furthermore, it is reasonable to expect that the visual stimulus

heavily modulates the statistics of the stochastic extra-areal input, making an approximation

with a stationary Poissonian input inappropriate. This latter issue is particularly problematic

given that we wished to study the modulation of fine temporal statistics in the spontaneous vs.

visual driven conditions.

To explore quantitatively the latter issue, we have repeated the experiment performed in section

4.4 that compares the trial-to-trial variability of the membrane potential in response to drifting

sinusoidal grating vs. naturalistic stimulus (Fig 10), but this time in a model enriched with an

external Poissonian input impinging on all cortical neurons at a rate of 1000 Hz (Fig S2C), which

corresponds to a roughly 1 Hz spontaneous rate arriving via 1000 extra-areal connections. As

can be seen, unlike in the original model (Fig S2A), the model with external input shows virtually

no change of trial-to-trial variability for both the grating and naturalistic conditions in

contradiction to the data (Fig S2D). This is because the constant, stimulus independent,

external Poissonian noise drowns the stimulus dependent modulation of variability within the

cortico-thalamic network.

For all these reasons our decision was to not include such compensatory external inputs in our

model. Yet, as we show in section 4.1 and 4.2, our model reaches realistic levels of

spontaneous and evoked activity at both the spiking, membrane potential and conductance

levels. How is this possible? As we explain in section 6.1.6, the existing evidence on the

synaptic strength of the various projections included in our model is very limited, leaving a lot of

leeway for setting the exact synaptic weights. To deal with this unknown we set the synaptic

weights to achieve an overall balance between excitation and inhibition that supports reasonable levels of both spontaneous and evoked activity, while being compatible with the

limited physiological findings. It is thus possible that these broad constraints on synaptic weights

allowed us to compensate for some of the external drive into stronger-than-biological synaptic

weights. This points to a general limitation of reductionist modeling where missing aspects of

the model are often (at least partially) absorbed into unrelated free parameters of the model.”

Thank you very much for extensively addressing the point raised in my remark. The provided answer is measured and satisfactory. One remaining request: Could you please add labels to the vertical axis in Fig 2S?

################################################################################

“l. 80: "Due to the low proportion of synapses of thalamic origin in the model cortex, the variability

of spontaneous and visually evoked cortical activity is largely driven by internal cortical

dynamics."

This is asserted without evidence. Surely, since neurons in L2/3 do not receive external input

from outside the model this is true trivially. But for neurons in L4 one can do the following

back-of-the-envelope calculation:

In the model, L4 excitatory cells have 1000 recurrent and around 140 TC synapses. Let's

assume that 50% of synapses are excitatory, 50% inhibitory (probably there are more

excitatory).

Spontaneous firing rate for excitatory neurons in L4 is 1.4 Hz, in L2/3 2 Hz. The weight for

L4E->L4E connections is 0.18 nS, for L2/3E->L4E 1 nS.

The latter connections are 20% of the synapses formed there, so 200 per cell.

This implies that the incoming recurrent input per excitatory neuron from all excitatory neurons is

about (omitting units) 500 * 0.8 * 1.4 * 0.18 + 500 * 0.2 * 2 * 1 = 100.8 + 200 = 300.

The inhibitory firing rate is 7.6 spikes/s in L4, 4.7 spikes/s in L2/3. The inhibitory weights are set

to 1 nS.

This results in incoming recurrent input per excitatory neuron from inhibitory neurons (again

omitting units, ignoring different signs from excitatory and inhibitory neurons) 500 * 0.8 * 7.6 * 1

+ 500 * 0.2 * 4.7 * 1 = 3040 + 470 = 3500.

There are 140 TC synapses per neuron firing at a rate of about 12 spikes/s with a weight of 1.2

nS leading to an input of about 140 * 12 * 1.2 = 2016.

The fraction of TC input to excitatory cells now is 2016 / (3500 + 300 + 2016) = 0.35.

This fraction gets higher if there are relatively more excitatory recurrent connections, so that it

may actually exceed 0.5. Furthermore, synchronization plays a role in determining which

connections are more influential. It is thus unclear if indeed "variability of spontaneous and

visually evoked activity is largely driven by internal cortical dynamics."

It is indeed correct that the thalamo-cortical drive plays a non-negligible role in the dynamics of

Layer 4 despite the low number of thalamo-cortical connections, but what we were emphasizing

here is that the internal cortical connectivity has a higher impact. We are thankful and impressed

that you took time to make these calculations, but your argumentation, as detailed as it is,

doesn’t take into account several other factors such as inhibitory synapses having a longer time

constant than excitatory synapses or that the thalamo-cortical synapses display more short-term

depression than the other synapses of the model due to their larger recovery time constant.

We believe that looking at the average synaptic conductances of the neurons in a version of the

model without any intra-cortical connections and comparing it to the average synaptic conductances in the Layer 4 neurons of the full model would be a better indicator of the impact

of the thalamo-cortical connections in the model’s dynamics. This measure doesn’t take into

account the synchronization of the inputs, but this factor is minimised for spontaneous activity

as the input to every LGN cell is independent noise, and cortical neurons have a low synchrony

in this condition. In response to drifting gratings, the average synaptic conductances of the

neurons selective to the orientation of the grating fall down to 20% of their original value when

we disable every cortico-cortical connection. For spontaneous activity, this value reaches 30%.

This supports us in stating that the activity of the model is largely driven by internal cortical

dynamics.

We have now added this figure as supplementary figure to the paper, and refer to it in the

statement claiming cortical dominance:

“Due to the low proportion of synapses of thalamic origin in the model cortex (Da Costa et al.

2011) the variability of spontaneous and visually evoked cortical activity is largely driven by

internal cortical dynamics (Figure S1).””

I appreciate the detailed reply. Indeed, as you note correctly, my calculations rested on some simplifying but eventually incorrect assumptions. I am however very happy that these motivated a numerical check of the claim.

The newly added figure (Figure S1) very nicely shows that indeed your claim can be justified.

################################################################################

“ "While most excitatory neurons in both cortical layers show surround suppression [...]."

This seems not to be quantified. In the caption of Figure 9 it says that traces of typical cells in L4

and L2/3 are shown. How many cells are "typical"?

We believe that this is quantified in the panel D of Figure 9, which shows the suppression index

of cells from both cortical layers. This panel shows that in majority excitatory cells in the model

have a non-zero suppression index which indicates that they display surround-suppression. It

also shows that Layer 4 cells can have different size-tuning profiles ranging from no suppression

to moderate suppression, as illustrated by the two Layer 4 example cells in panel A. Finally, this

panel highlights that the vast majority of Layer 2/3 excitatory cells display moderate

surround-suppression as illustrated by the Layer 2/3 example cell in panel A.”

Sorry for being imprecise. What I meant is that I would have liked to see the fraction of cells with a SI greater than 0 or another threshold value. I think giving this value does not make your point weaker since to my understanding your results in that regard are compatible with the ones presented in Tailby et al.

################################################################################

“We observed that the addition of this neuron-type-dependent delay factor improved the

stability of the modeled cortical neural networks, reducing synchronous events during

spontaneous activity."

This is not a new observation, see for example Brunel 2000.

This is not a new observation, see for example Brunel 2000.

We thank you for notifying us that other studies observed the same phenomenon. We did not

claim here that we have discovered something new, we are just stating our observation, and we

would be happy to reference a study that would support it. Nevertheless, after reading Brunel

2000, we understand that in this case it was not the introduction of different constant

transmission delays for the different type of connections that enhanced the stability of the

asynchronous state, but rather the sampling of each transmission delays from wide

distributions, as shown by the following sentences in their subsection “5.2. Wide Distribution of

Delays”:

“The analysis of the previous section can be generalized to the case in which delays are no

longer fixed but are rather drawn randomly and independently at each synaptic site from a

distribution Pr(D)

[...]

Thus a wide distribution of delays has a rather drastic effect on the phase diagram, by greatly

expanding the region of stability of the asynchronous stationary state. The other regions that

survive are qualitatively unmodified.”

Later in the article, the author describes a second model (Model B) for which excitatory and

inhibitory connections have different characteristics, and notably different transmission delays

distributions across E-E, E-I, I-E, and I-I connections, but he doesn’t report that this specific

change has increased the stability of the asynchronous state of the network.

We therefore chose not to reference this study in our manuscript.”

Thank you for addressing this so carefully. You are absolutely correct, in Brunel 2000 the author addresses the case where all delays are sampled from a random distribution. I misunderstood your sentence initially, but it is indeed sufficiently clear.

################################################################################

New comments:

Could you please clarify in the caption of Table 2 whether the rows are pre- or postsynaptic populations (and similarly for the columns)? One can get it from the context but it still would increase readability.

Table S1:

You write under topology:

“Square 2D sheet using cortical distance” – To my understanding you use the Euclidean distance, I think “cortical distance” might be a bit confusing here.

Reviewer #2: Attached (very brief replies in blue thanking the authors for their updates and attention to my comments)

Reviewer #3: The authors have addressed all my queries in a satisfactory manner.

**Have the authors made all data and (if applicable) computational code underlying the findings in their manuscript fully available?**

Reviewer #1: Yes

Reviewer #2: Yes

Reviewer #3: Yes

PLOS authors have the option to publish the peer review history of their article (what does this mean?). If published, this will include your full peer review and any attached files.

Reviewer #1: No

Reviewer #2: No

Reviewer #3: No

Figure Files:

Data Requirements:

Reproducibility:

References:

---

## [Editor Report · Decision Letter 2]

20 Jul 2024

Dear Antolik,

We are pleased to inform you that your manuscript 'A comprehensive data-driven model of cat primary visual cortex' has been provisionally accepted for publication in PLOS Computational Biology.

Best regards,

Hugues Berry

Section Editor

PLOS Computational Biology

Hugues Berry

Section Editor

PLOS Computational Biology

---

## [Editor Report · Acceptance letter]

15 Aug 2024

PCOMPBIOL-D-23-01929R2 

A comprehensive data-driven model of cat primary visual cortex

Dear Dr Antolík,

I am pleased to inform you that your manuscript has been formally accepted for publication in PLOS Computational Biology. Your manuscript is now with our production department and you will be notified of the publication date in due course.

With kind regards,

Lilla Horvath
